# Polymeric and Paper-Based Lab-on-a-Chip Devices in Food Safety: A Review

**DOI:** 10.3390/mi14050986

**Published:** 2023-04-30

**Authors:** Athina-Marina Mitrogiannopoulou, Vasiliki Tselepi, Kosmas Ellinas

**Affiliations:** Department of Food Science and Nutrition, School of the Environment, University of the Aegean, Ierou Lochou & Makrygianni St, GR 81400 Myrina, Greece

**Keywords:** lab on a chip, microfluidics, polymer-based, paper-based, fabrication techniques, pathogen detection, food safety, water safety

## Abstract

Food quality and safety are important to protect consumers from foodborne illnesses. Currently, laboratory scale analysis, which takes several days to complete, is the main way to ensure the absence of pathogenic microorganisms in a wide range of food products. However, new methods such as PCR, ELISA, or even accelerated plate culture tests have been proposed for the rapid detection of pathogens. Lab-on-chip (LOC) devices and microfluidics are miniaturized devices that can enable faster, easier, and at the point of interest analysis. Nowadays, methods such as PCR are often coupled with microfluidics, providing new LOC devices that can replace or complement the standard methods by offering highly sensitive, fast, and on-site analysis. This review’s objective is to present an overview of recent advances in LOCs used for the identification of the most prevalent foodborne and waterborne pathogens that put consumer health at risk. In particular, the paper is organized as follows: first, we discuss the main fabrication methods of microfluidics as well as the most popular materials used, and then we present recent literature examples for LOCs used for the detection of pathogenic bacteria found in water and other food samples. In the final section, we summarize our findings and also provide our point of view on the challenges and opportunities in the field.

## 1. Introduction

Lab-on-a-chip devices (LOCs) are miniaturized devices in which biological, chemical, or biochemical analyses that are normally conducted in a laboratory are performed inside a single device [1]. Such devices exhibit several advantages, such as the small amount of sample required, portability, low price compared to other methods, and fast and accurate results. In addition, they do not require a centralized laboratory with dedicated equipment and highly trained personnel [2,3]. Due to these advantages, LOCs can find a wide range of applications in healthcare [4], food safety [5], and chemical analysis [6] and have become a highly interesting research area. However, how was this idea developed? The idea of LOC is based on microfluidics, which is directly related to the development of semiconductors and the use of micro and nanotechnology. The first micro-sized transistors were fabricated using photolithography in 1950 [7] and marked the beginning of microelectronics and Micro-Electromechanical Systems (MEMS). The first Lab-on-a-chip device was reported in 1979 by Stanford University [8]. However, the first proponent of “Lab-on-a-chip” devices is considered to be Andreas Manz, who created the definition for μTAS (micrometer-scale total analysis systems), “a system that periodically performs ALL sample handling steps required to translate chemical into electronic information at a location that is extremely close to the point of sample collection” [9,10].

Similarly to the μTAS definition, LOCs are devices in which all necessary steps during a sample analysis can be completed in a single device. Basic functions such as sample preparation, mixing, filtration, heating, separation, and detection can be integrated into a LOC operation [11]. It is therefore evident that LOCs are not simple arrangements of microchannels but rather a more sophisticated assembly of various parts that enable specific functions. A complete and autonomously operating LOC may require the integration of several parts, such as micro-pumps, micro-valves, micro-mixers, reaction chambers, actuators, and sensors. More interestingly, parts such as the micro-valves, which control the fluid flow inside the microchannels, can be operated manually, pneumatically, or electrokinetically [12,13]. A representative example of a LOC for biological analysis is provided in Figure 1A, while Figure 1B shows the most commonly used materials, fabrication methods, and detection principles in microfluidics.

In terms of materials used in LOC fabrication, silicon and glass were the first materials used since the first introduction of μTAS in 1990 [9], but interest quickly switched to other, less expensive, and easier-to-handle and manipulate materials like elastomers, thermoplastics, and paper [16].

As stated above, LOC devices have been used in various fields, such as chemistry, drug production, medicine, proteomics, and food safety, and their usage has only been expanding during the past few years [17,18,19,20]. In developing countries, their application in the healthcare field can be characterized as urgent and inevitable, with over 1 billion people lacking basic health care services [4,21]. Food and water safety is another key area of application for LOCs. Again, many people in developing countries, i.e., in the Middle East, Asia, and North Africa, continue to lack access to basic essentials such as clean drinking water and healthy food, despite all the technological breakthroughs [22]. However, it is not only a need in developing countries, food safety is an important field worldwide, and it is estimated that over 100 million people are annually affected by Food Borne Diseases (FBD) with 32 million being children under 5 years old. Pathogen microorganisms, especially intestinal bacteria, such as *Salmonella*, *Campylobacter*, *E. coli* O157:H7, and *Listeria* monocytogenes, are mainly responsible for food and water deterioration and are causing most of the foodborne illnesses.

Thus, the application of LOC devices in food safety-related applications is blooming since LOCs can provide rapid, accurate, and on-site analysis of food samples. In this review paper, we provide an overview of the main fabrication techniques and the applications of microfluidic devices, as well as a complete literature review of efforts about polymeric and paper-based LOCs used for detection of food-borne and water-borne pathogens. In the final section, we summarize the most important advances in the field together with their advantages and disadvantages, and we present our opinion on the remaining tasks required in order to accelerate the industry’s uptake of this new type of devices.

## 2. Polymer and Paper-Based Microfluidic Fabrication Methods

Since the introduction of the μTAS concept in the 1990s, scientists have gradually shifted their interest in polymers due to the demand for cheap and disposable devices. Polymer materials typically used in microfluidic applications can be divided into three categories according to their physical properties: (a) thermosets, (b) thermoplastics, and (c) elastomers [23].

Thermosets are polymers that are irreversibly hardened and come from soft, solid, or viscous liquid prepolymers. Curing results in chemical reactions that create extensive cross-linking between polymer chains to produce an infusible, insoluble, and thermally stable polymer network [24]. Examples of thermosets are polyimide, SU-8 (photoresist), polyurethane, and epoxy resin.

Thermoplastics, on the other hand, do not form irreversible chemical bonds during the curing process because their polymer chains are not cross-linked, so they can move inside the bulk at elevated temperatures, which is the reason why they are deformable [25]. They are typically used to produce parts using various polymer processing techniques such as injection molding, compression molding, or hot embossing. Examples of thermoplastic polymers that are commonly used in microfluidics applications are polystyrene (PS), polycarbonate (PC), and Cyclic Olefin Copolymer (COC) [26].

Elastomers consist of cross-linked polymer chains that can be deformed under pressure but come back to their initial shape when the force stops. The most common elastomer is PDMS (polydimethylsiloxane) which belongs to the group of silicon elastomers and can be deformed under force or air pressure [27]. PDMS is easy to work with, and its microfabrication is not costly. Liquid PDMS prepolymer is cured at temperatures between 40 °C–70 °C and it can be cast in nanometer resolution from photoresist templates that are easier and cheaper to prepare than silicon or glass. It also has great biocompatibility and biostability and is used in many medical applications [28].

Paper is another material commonly used in microfluidics and LOCs; it is produced from a dilute aqueous suspension of cellulose fibers that are drained through a sieve, then pressed and dried to yield a sheet formed by a network of randomly interwoven fibers [29]. Its unique structure provides essential advantages for microfluidics applications: (a) capillary action, allowing liquid samples to be directed along the desired path without external forces [30], (b) good reagent distribution due to the high absorbency and air permeability [31], (c) sample filtration through the network structure of the paper, and (d) a high surface-to-volume ratio, which increases the number of possible immobilization sites [30] (e) natural biocompatibility, biodegradability, and chemical and biological inertness. A landmark report from the group of Whitesides in 2007 on the use of paper patterned with photoresist to create channels and colorimetric detection zones initiated an avalanche of work in the field due to the multiple advantages of paper as well as its availability all around the world [31]. In the following sections, we provide an overview of the current fabrication techniques for polymeric as well as paper-based microfluidics and LOCs, focusing mainly on examples related to food safety applications.

### 2.1. Polymer Mold-Based Techniques

Thanks to their properties, polymers can be easily used as base materials for several parts, objects, and components in many industry sectors. The main production method for plastic objects is molding. In microfluidics, molding polymers on the microscale is one of the most common techniques for the realization of microfluidics and LOCs. In the following sections, we provide examples of the different molding techniques used in microfluidics.

#### 2.1.1. Hot Embossing

Hot embossing is a technique that uses the very simple concept of transferring patterns of micro- and nanoscale features from a mold to a target substrate and has been extensively used for the fabrication of various microfluidic devices. In the past two decades, this technique has successfully broken through the barrier of laboratory-scale production and become an industrial-scale production technique [32]. Overall, there are two types of hot embossing: the conventional method (plate-to-plate) [33] and roll-to-roll [34]. Using this approach, the productivity of the method is greatly improved. A schematic representation of both methods is provided in Figure 2.

Several examples of microfluidics or LOCs fabricated using hot embossing exist in the literature, such as the fabrication of a microfluidic platform for ELISA on beads targeting the detection of two important human proteins (IL6 & PDGF-2). The device included superhydrophobic inlet and outlet valves as well as an incubation chamber, and it was created by hot embossing the Si master onto a PMMA substrate [35]. Similarly, Kourmpetis et al. [36] produced microfluidic chromatography columns by hot embossing, but in this approach, they held the two plates (one plate with the Si master and the other with the COP substrate) at different temperatures in order to avoid deformation as well as to achieve a successful replication of fine micropillars. Studies related to hot embossing include several review papers in which all parameters affecting the quality of the replication are analyzed in detail [37,38].

Food-relevant efforts include the work by Matlock-Colangelo et al. [39], who fabricated a PMMA-based LOC that included positively and negatively charged electrospun poly (vinyl alcohol) (PVA) nanofibers to detect *E. coli* K12 cells. The negatively charged nanofiber mats managed to capture 72% of the *E. coli* cells while reducing nonspecific analyte retention within the channel. The microfluidic channels (four parallel channels, 42 μm deep, 1 mm wide, and 20 mm long each) were embossed into pieces of PMMA using a Hot Press at 130 °C and 10,000 lb (44,482 N) of force for 5 min. Furthermore, UVO-assisted thermal bonding was used to produce microfluidic devices. In another example, a deep (28  μm) cyclo-olefin polymer (COP) microcolumn with high aspect ratio (3:1) features was fabricated using a two-step process: (a) hot embossing of a-Si master and (b) pressure-assisted thermal bonding. The column was used to separate substances used in the food industry such as sodium benzoate, methylparaben, and propylparaben. To reduce processing time, techniques such as ultrasonic hot embossing have also been developed [38,40], in which the embossing action is created by pressing a sonotrode vibrating at an ultrasonic frequency against the mold, causing localized heat generation that softens the plastic substrate, and the pattern is transferred from the mold [41]. In another work, far-infrared (FIR)-based heating was combined with hot embossing, and the substrate temperature rose from room temperature to the embossing temperature of 135 °C in just 25 s, while the cooling was accomplished in 90 s by the circulation of water through the cooling channel, so the total process cycle time was less than 2 min [42].

#### 2.1.2. Injection Molding

Injection molding is a manufacturing process in which molten material is injected into a heated mold, and after curing, the patterned parts are produced. It is considered one of the most effective methods for the mass production of thermoplastic polymer micro-components. For conventional injection molding, plastic pellets or granules are fed from a hopper into a heating barrel, and a screw-type plunger moves the material slowly forward as it melts [43]. Heating can be achieved by hot air, coils, water, steam, heating cartridges, heating rods [44], or by exploiting the radiation heating from sources such as infrared lights, lasers, and the proximity effect of a high-frequency current through an electrical coil. In conventional injection molding, the mold is kept below or equal to the polymer’s glass transition temperature. However, there are also non-conventional injection molding methods, such as the variothermal process, in which the mold temperature is higher or equal to the polymer’s glass transition temperature during injection and changes rapidly during cooling. The cycles of the variothermal process can be modified to match the desired conditions and achieve higher quality [45]. Another method considered non-conventional is ultrasonically assisted injection molding. This method has several benefits over conventional microinjection processes, with the most important being the reduced energy required. However, according to Whitesides et al. [46], there are concerns about whether such systems can produce homogeneous melting. A schematic representation of a typical injection molding process is provided in Figure 3.

Fabricating microfluidic chips with injection molding is a process that requires processing molds, which is a costly and time-consuming process. Hansen et al. [47] followed an approach of depositing SU-8 photoresist directly on the surface of nickel molds so they could be reused 300 times with cyclic olefin copolymer (COC) without any signs of failure or release and, in that way, reduced the cost and the time needed for fabrication. In the food industry, injection molding has been used by Yu et al. [48] as the fabrication process for a centrifugal three-layered microfluidic chip from PMMA that contains reaction cells connected to a buffer cell, which in turn is connected to the main road, inlets, and outlets. This chip uses a real-time fluorescent LAMP method for detecting samples simultaneously for five types of milk (cow, goat, horse, camel, and yak). All the above highlights that injection molding is a technique that has very good repeatability, is fast, can process 3D microfluidic chips, and can be used for large-scale production, while the main drawback of the method is related to the high cost of the mold.

#### 2.1.3. Casting

Casting is a manufacturing process in which a liquid material is poured into a mold that has the desired shape and then is left to solidify. Casting is considered a low-cost and simple procedure with high fidelity. The most commonly used material in this process is Poly(dimethylsiloxane) PDMS, and the techniques derived from using it, are termed “soft lithography”, which was initially developed by Whitesides et al. in the late 1990s [49]. The elastomer pouring technique was developed by Bell Laboratory in the 1970s [50] and was first used in the 1980s to fabricate microfluidic chips. For example, in 2021, Xue et al. [51] used the casting method for the PDMS mold preparation in the fabrication process of a microfluidic chip for the detection of foodborne *Salmonella*. Furthermore, Salih et al. [52] detected coliform bacteria in water by measuring the absorbance of the sample using UV-visible spectroscopy, with a detection limit of 17,200 cfu/mL. Their portable device was fabricated with glass and polydimethylsiloxane (PDMS) material using replica molding/casting (soft lithography) to create the PDMS microchannel. Figure 4 includes a schematic representation of the mold and its application to create a PDMS-based microfluidic.

### 2.2. Micromachining Techniques

CNC machining is a manufacturing process in which factory tools and machinery are manipulated by computer software. The method’s versatility and simplicity have made it also popular for the fabrication of microfluidic chips, but the resolution of the method is determined by the size of the available tools; up to now, microfluidics with channel dimensions greater than 100 μm have been reported [43,53].

Laser micromachining is a technique that involves other steps and processes such as laser milling, cutting, etching, and engraving. These processes are also known as laser ablation where a high-energy laser beam is used and focused on the substrate surface to remove materials for the design of micro/nanostructures. The laser sources that are used can be classified based on their wavelength (UV/excimer lasers and infrared lasers) or the time scale of their pulse durations (millisecond, microsecond, nanosecond, picosecond, and femtosecond lasers). The main advantages of this method are the high precision and the method flexibility. Using laser micromachining, a wide range of polymers have been used for the fabrication of microchannels. Examples include COP [54], PMMA [55], PS, PC [56], PET [57], PDMS [58], as well as biodegradable polymers [59]. One interesting example is the work by Bilican et al. [60], in which CO_2_ laser machining was used to create microchannels in PMMA and PS, followed by detailed thermal and material properties analysis, trying to define the capabilities as well as the limitations of the method.

### 2.3. 3D Printing

3D printing has also attracted a great deal of attention for the fabrication of microfluidics. 3D printing enables the fabrication of complex flow-regulating components and the integration of detectors and cell culture on chips. It can be carried out through processes such as stereolithography (SL) [61], selective laser melting and sintering (SLS) [62], fused deposition [63], and poly-jet or multi-jet modeling (MJM). For example, Duarte et al. [64] studied the development of 3D-printed microfluidic devices with integrated electrodes for label-free counting of *E. coli* cells incorporated inside droplets based on capacitively coupled contactless conductivity detection (C4D). The devices were fabricated with the use of a 3D printer with fused deposition modeling (FDM), which is one of the simplest and lowest-cost methods. It is based on the fabrication of 3D objects with layer-by-layer deposition of semi-fused thermoplastic filaments, such as polylactic acid (PLA) and acrylonitrile butadiene styrene (ABS), on a printing platform [65]. The group managed to detect *E. coli* cells in the concentration range between 86.5 and 8650 cfu/droplet, with a detection limit of 63.66 cfu/droplet. Kanitthamniyom et al. [66] describe the development of a magnetic digital microfluidic diagnostic platform for rapid, accurate, and parallelized solutions for clinical Carbapenemase-producing Enterobacteriaceae (CPE) detection. They isolated and tested 27 bacteria from the platform and observed the color change in the droplets, indicating the bacteria’s presence. Figure 5 exhibits how simply a CAD design can be transformed into a microfluidic device using the 3D printing approach.

### 2.4. Optical Lithography Techniques

Optical lithography-based methods have been extensively used for the fabrication of microfluidic channels. SU-8 is one of the most popular materials in this method as it has high mechanical rigidity, and chemical stability, as well as a defined shape and size [68]. SU-8 has also been used for the fabrication of masters of in-line micro-valves with different aspect ratios via lithography techniques due to its chemical and thermal stability after polymerization and allows mass production from a single master mold [69]. In another interesting work, polystyrene (PS), which is a commonly used material for biological and biomedical applications due to its high biocompatibility, has been used for the fabrication of microstructures with 20 μm resolution with direct optical lithography [70].

### 2.5. Plasma Processing

Plasma processing has been used in microfluidics to selectively modify the wetting properties of microfluidics [71], to promote the adhesion of a bio interface or a coating inside microfluidics [72], as well as for surface treatment to achieve bonding between glass and PDMS or other polymers [73]. However, its application is not limited to the aforementioned applications, and if combined with other methods such as optical lithography, it can provide an alternative method for the complete fabrication of a functional polymer-based microfluidic device [74].

For microfluidics relevant to food safety applications, plasma processing has been used to enable antibody binding at a high level, and microfluidic chips for bacteria capturing and lysis have been demonstrated on PMMA substrates. In particular, the proposed devices have been used for the detection of *Salmonella* with concentrations ranging from 10^2^ to 10^8^ cells/mL. Except for *Salmonella*, *E. coli*, and S. Typhimurium were also successfully captured and detected with a capture efficiency of 80–100% for concentrations of less than 10^6^ cells/mL even when using high flow rates [75]. In another example, Geissler et al. [76] fabricated a microfluidic chip for the identification of *E. coli* after a short analysis time. The device integrated thermal lysis, PCR amplification, and microarray hybridization on the same cartridge [77,78]. The immobilization of the amino-modified DNA probes was completed by exposure of the polymer materials to oxygen plasma, leading to the formation of hydrophilic, oxygen-containing species (including –OH groups) which increase the surface free energy and promote the wetting of the polymer substrate by polar solvents.

Table 1 summarizes the most common fabrication methods for polymeric microfluidics, together with their advantages and disadvantages. There is no “perfect” solution when choosing the fabrication method of a polymer-based microfluidic, and the choice is determined by several factors such as accessibility to a clean room, the material selection for the application envisioned, and most importantly, the final intended use of the device. For example, injection molding and roll-to-roll hot embossing are ideal for high-volume production, but the material undergoes wide changes in temperature. On the other hand, technologies such as plasma treatment seem really promising for the functionalization of such devices. The selection of the fabrication method also depends on the complexity of the device, the size and shape of the channels, and the material being used. Each technique has its own advantages and disadvantages. However, soft lithography is considered to be one of the most popular and versatile methods for the fabrication of polymer-based LOC devices for research purposes in food testing, as it is relatively simple, inexpensive, and reproducible. However, thermoplastic polymers are mainly used by companies working with microfluidics for commercial purposes due to the large-scale fabrication methods for thermoplastics (i.e., injection molding) and the properties (i.e., transparency, rigidity) such materials can offer.

### 2.6. Paper-Based Microfluidics

Microfluidic chips based on paper are called paper-based microfluidic analytic devices (μPADs) and since their development in the early 21st century, they are considered a rapidly growing field. They can be manufactured in 2D or 3D dimensions. For the fabrication of μPADs techniques such as inkjet printing, wax printing, photolithography, plasma treatment, laser treatment, etching, and 3D printing have been employed. 

Historically, Whitesides’ group started the field of μPAD devices but the first fabrication of a paper device that includes a defined fluidic channel was investigated by Muller and Clegg in 1949 [79]. The idea behind μPADs is to provide low-cost, disposable, simple-to-use, analytical devices that are about to be used in low-resource settings, such as developing countries, or for on-site analysis in which technical infrastructure or trained personnel are limited or absent [80]. μPADs can be fabricated by either patterning a hydrophobic barrier on the paper substrate [81,82,83] or shaping/cutting the paper to define the fluidic channels [82,84,85,86]. In the next sections, we provide an overview of some widely used fabrication techniques for paper-based microfluidics, while in Figure 6 some important milestones in the use of paper-based kits are presented, starting with the invention of text paper in the 17th century.

#### 2.6.1. Wax Printing

Wax printing is an easy-to-implement and inexpensive technique that can be utilized even by researchers with a limited budget or no prior micro-fabrication experience [88]. Wax printing involves a two-step protocol: wax is printed on the paper surface, and then it is melted to form hydrophobic barriers. The fabrication time of devices with this technique can be less than 5 min (from design to finished prototype) [89]. To avoid the use of dedicated wax printing on commercial printers and cartridges, researchers have used toner or laser printers [90].

Asif et al. [91] investigated μPADS for the detection of *Staphylococcus aureus* (*S. aureus*) and *E. coli* in milk samples. They printed an array of 7 mm diameter circles with a 0.5 mm line thickness on paper using a wax printer. After that, it was impregnated with chromogenic substrates, which react with bacterial enzymes, providing a clear color change that was monitored by UV-vis spectrophotometric analysis. The detection limits for *S. aureus* and *E. coli* were found to be 10^6^ cfu/mL, but after enrichment of the milk samples in a selective medium for 12 h detection of samples containing as low as 10 cfu/mL was possible. The devices were tested on a set of 640 milk samples collected from dairy animals in Pakistan and demonstrated more than 90% sensitivity and 100% selectivity when compared to PCR. In another application, Zhao et al. [92] used artificially contaminated beef samples to test a wax-printed paper-based enzyme-linked immunosorbent assay (P-ELISA) with two monoclonal antibodies against *E. coli* O157:H7 with a test time of 3 h and 5 μL of the sample. With the use of the wax printer, 96-well plates were printed on paper and then heated for the wax to melt and form the barriers. Six concentrations of bacteria, ranging from 10^3^–10^8^ cfu/mL were used, and parallel experiments were conducted. This group came to the conclusion that the P-ELISA method was faster, less costly, more sensitive, and more specific than other methods such as C-ELISA and PCR. A schematic representation of the wax printing procedure is presented in Figure 7.

#### 2.6.2. Inkjet Printing

Inkjet printing is a simple and useful technology for the accurate and contactless dispensing of picolitre-sized droplets of liquids (inks) onto a user-defined position on a substrate [80]. In microfluidic channels, it has been used to deposit hydrophobic materials as barriers; commonly used materials are polystyrene [85] (Figure 8), hydrophobic sol-gel [94], silicone [95], and alkyl ketene dimer [96]. Inkjet printing is a highly adapted method for material deposition, and several examples of its application exist in the literature [97,98,99,100]. One example in which inkjet printing was used is the work by Hossain et al. [101] for the detection of food- and water-borne bacteria (*E. coli* O157:H7 and *E. coli* O104:H4). Snyder et al. [102] reduced the analysis time for the detection of coliforms from 24 h to 6 h with a paper-based device that performs cell lysing on-chip, at an initial *E. coli* concentration of 1 cfu/mL after incubation. Without any incubation, the device could detect bacterial concentrations as low as∼10^4^ cfu/mL. The bacteria *E. coli* were detected via the presence of the coliform-specific agent, β-galactosidase.

#### 2.6.3. Optical Lithography

Optical lithography has been used to create high-resolution features for paper-based devices with sizes down to 100 μm and hydrophobic barriers as small as 200 μm in width. The photoresists that are used the most in photolithography are poly (o-nitrobenzyl methacrylate) (PoNBMA) [104], octadecyl trichlorosilane (OTS) [105], and SU-8, with which Whitesides et al. [31] prepared and designed the first μPAD. However, photoresist is the major cost in the photolithography process, with SU-8 being costly enough (e.g., SU-8 costs 1 $/g and PMMA 0.15 $/g [103]), so with that in mind, Whitesides replaced SU-8 with cyclized poly (isoprene) derivative photoresist [106] and propanediol methyl ether acetate (PGMEA).

Several examples of paper-based devices for the detection of foodborne pathogens exist in the literature, mainly due to their simple and cheap fabrication process. Lin et al. [107] designed a paper-based analytical device for the detection of *E. coli*, in both tap water and seawater samples. For the fabrication of the device, the filter paper was immersed in a mixed solution of water-based PUA (polyurethane acrylate), which is a photosensitive resin, and a photoinitiator (HMPP- 2-hydroxy-2-methylpropiophenone). After absorption and baking, UV photolithography was used to pattern the paper-based device. For the detection of *E. coli*. 10 μL of bacteria lysate was dropped onto the colorimetric substrate, and after 10 min the color intensity was measured with an in-house-made device that includes an integrated cadmium column with a detection limit of 3.7 × 10^3^ cfu/mL [108]. Bacteria samples with concentrations ranging from 10^1^ to 10^9^ cfu/mL were investigated, and the color change from yellow to red/violet was measured when the intracellular enzyme β-galactosidase reacted with a chromogenic substrate, CPRG (chlorophenols). The intensity of the color here depends linearly upon the concentration of *E. coli*. and corresponds to a range of 10^4^–10^9^ cfu/mL. For tap water, the detection range was ~10^5^–10^6^ cfu/mL as well as for seawater ~10^4^–10^6^ cfu/mL. Photolithography can be combined with other fabrication techniques in order to produce the desired microfluidic chip. An interesting example of such a combination is the work by Yu et al. [109], who used photolithography with UV photosensitive inks on a piece of Parafilm to pattern the microchannels and wells and then embossed the Parafilm on Whatman #1 filter paper in an oven, to transfer the layout. A typical fabrication process flow is shown in Figure 9.

#### 2.6.4. Screen Printing

Screen printing is a traditional printing technique that involves a customized mesh that is used to selectively transfer ink or dye onto a substrate. Although screen printing can be used to produce many devices, it requires a customized screen via photolithography to transfer the pattern onto a substrate. Each screen, even though it is reusable, is usually customized; therefore, screen printing is a time-consuming and costly process. Furthermore, such devices’ feature sizes and spatial resolution are high, e.g., in the millimeter-scale range. Rengaraj et al. [110] developed a device consisting of paper-based electrodes for the impedimetric detection of bacteria in water. The sensing probe was fabricated by screen printing three layers of conductive carbon-based ink onto a commercial hydrophobic paper, and the electrode surface was modified with carboxyl groups before prevalent immobilization of the lectin Concanavalin A (Con A), which was used as the biorecognition element. The three-layer printing methodology was selected as the best choice between low resistance (18 Ω/cm) and rapid printing. The final device consisted of a circular working electrode with a 6 mm diameter and a geometric surface area of 0.286 cm^2^, printed on a paper strip with a 4 cm length and 1 cm width to cross-link the cellulose fibers; after screen printing the electrodes, the paper was submerged for 3 h in a solution of 6% *w/v* glyoxal and then thermally treated. The bacterial detection limit obtained was 1.9 × 10^3^ cfu/mL and the dynamic range was 10^3^–10^6^ cfu/mL.

#### 2.6.5. Laser Processing Technology

Among the physical fabrication processes, some are complex, such as inkjet etching, and some offer low fabrication resolution, such as wax patterning. Laser processing is the laser beam irradiation that removes or melts the material and changes the surface properties of the object, by exploiting the laser’s high energy. In particular, in μPAD fabrication, the laser is used to remove the hydrophilic areas and create a hydrophobic barrier. Mahmud et al. [111] developed a fabrication technique for patterning compact and microscale features with a laser cutting machine on chromatography paper backed with aluminum foil. In that way, the μPADs that were manufactured can have small sample volumes, small chemical reagent volumes, and a reduced packing cost. They created channel barriers with a width of 39 ± 15 µm that were capable of restricting fluid flow across the barrier, thus generating channels with a width of 128 ± 30 µm. In another study, Bagheri et al. [112] cut a Whatman filter paper No.1 with a CO_2_ laser engraver to form a Y-sign design with 600 μm wide channels and, in that way, shape hydrophobic barriers. To prevent defects from burning, the paper was immersed in distilled water for 1 day after the cut. *S. aureus* was detected in food samples of orange juice and milk with a colorimetric aptamer-based Au/Pt NCs sensor at concentrations of 10^2^–10^8^ cfu/mL and with a detection limit of 80 cfu/mL. As the laser-matter interaction is a complex phenomenon, laser processing applications require accurate mathematical models to describe various processes that occurr during the laser interaction with different materials, as with paper [113].

#### 2.6.6. Plasma Processing

Plasma treatment is also used in paper-based microfluidics and aims to modify the chemical and physical properties of a surface. Plasma is used to fabricate hydrophilic patterns on paper samples, as in the study of Li et al. [114] that used Whatman filter paper as a substrate and hydrophobized it with the use of an alkyl ketene dimer (AKD) heptane (0.6 g/L) solution. The plasma-treated areas were strongly wettable by water or other aqueous solutions and allowed the transport of liquids along and within the plasma-treated channels through capillary penetration. In order to test the device, they observed a color change reaction for the evaluation of enzyme activity, so they came to the conclusion that this fabrication method can have desirable results in creating a microfluidic chip that can transfer and analyze samples. A usual problem with plasma treatment is the over-etching effect, which has been observed many times. Over-etching happens because the particles generated in vacuum plasma have long mean free paths, and this causes the treated pattern to be slightly bigger than the mask. However, it can be controlled by controlling the process parameters.

#### 2.6.7. 3D Printing/Lamination Methods

3D printing involves the layer-by-layer printing of an integrated device that has been designed using computer-aided design (CAD) software [115]. There are 3D printing techniques that are used for industrial as well as commercial purposes, such as Stereolithography (SLA) [116], Fused Deposition Method (FDM) [117], or the upcoming Electron Beam Melting (EBM) [118], and Bioprinters [119,120]. In stereolithography, consecutive layers of photocurable resin are laid, and a UV laser is used to cure the material. Fu et al. [121] adopted digital light processing stereolithography (DLP-SLA) 3D printing technology to fabricate 3D-μPADs, in which the interlayer bonding and alignment were automated and not assembled manually as in stacking or folding methods. For the fabrication of the μPADs print, files were made in advance, and the paper was immersed in the resin tank to be able to automatically bond in the layer-by-layer procedure. Furthermore, the fluid flow was triggered by an electric field or airflow, and a colorimetric assay was used to check the flow controllability.

In laminate manufacturing, individual layers are fabricated and then joined together in a stack to create the final microfluidic device. Different levels of channel structures are created on multiple sheets of paper with the same shape and size, and then a double-sided tape, a clip, or any other device is used to fix them into a whole paper chip. Stacking two or more layers of patterned paper on top of each other creates a three-dimensional microfluidic device in which fluids can move in all directions. The advantage of this type of paper-based microfluidics is that the density of channels increases, so more complex fluid handling processes can take place, contrary to the 2D schemes [122,123].

In Table 2, we summarize the main fabrication methods for paper-based microfluidics, along with their strengths and weaknesses. Similarly, for polymer-based devices, the choice of the method as well as the materials should be completed with respect to the requirements of the devices in terms of resolution, productivity, and the applications envisioned. Some of the most commonly used techniques in food testing are photolithography, wax printing, and inkjet printing. Each technique aims to create hydrophobic physical barriers on hydrophilic paper to passively transfer aqueous solutions. Again, we can find high-throughput methods that are less accurate and others that are more accurate but are only suitable for small-scale production or just prototyping. Photolithography, for example, is a high-precision technique that can produce complex designs with high resolution, but it requires a clean room, while wax printing is probably the cheapest technique, but it has limited resolution. Inkjet printing is a versatile and low-cost technique that can print different types of liquids and biomolecules directly onto paper, but it requires careful selection of the ink and paper and can have poor reproducibility.

As mentioned above, the paper used in paper-based LOC devices is typically a type of cellulose-based paper that has been modified to have specific properties. In particular, the paper should have a high wicking rate, which means that it can quickly and uniformly distribute liquid, and a low autofluorescence, which minimizes background signal in detection methods such as fluorescence. The choice of paper can also depend on other factors, such as cost, availability, and compatibility with other components of the LOC device. For example, some applications may require a paper that is chemically resistant, mechanically robust, or has specific surface properties for cell adhesion or capture. Overall, the choice of paper material will depend on the specific requirements of the application and the type of fabrication method being used.

## 3. Microfluidics and LOCs in Food Safety Applications

Food and water are crucial domains that necessitate the development of innovative diagnostic tools since the prevalence of food-related disorders and poisoning outbreaks increases globally [124]. Standard microbiological techniques (i.e., plate culturing) require several days for the final results [125]. This lengthy lag between sample collection and analysis hinders prompt measures to prevent outbreaks. According to Regulation (EC) No 178/2002 of the European Parliament and the Council laying down the general principles and conditions for food-related laws, water is considered “food.” Infections transmitted by food and water are typically caused by bacteria [126] in raw foods, and their presence, even in minute quantities, can cause diseases. For the reason that agricultural, breeding, and, in general, food production methods have changed to accommodate the ever-increasing global population, food quality is deteriorating. Pesticides [127], heavy metals [128], and pathogenic bacteria [129] have been detected in various stages, from production through processing, storage, and eventually consumption. Therefore, the products must be subjected to rapid quality control. Considering this, the demand for LOC devices that can conduct tests quickly and with a small number of required samples is crucial.

Biosensors play a key role in the development of LOC devices since they can identify biological elements, such as microorganisms, cells, tissues, and enzymes, and translate them into different signals (e.g., optical, electrical, or acoustic) [130]. For the majority of tests performed with LOC devices, no pretreatment steps are required because the sample is processed on the microchip, where operations such as amplification of the DNA and cell lysis occur [131]. The basic components of a biosensor are shown in Figure 10.

Several examples of pathogenic microorganism detection in food via microfluidic devices will be provided in the following section. As far as water is concerned, one of the most dangerous bacteria in water that poses a risk to human health is the Legionella bacterium. Legionella, whose most dangerous strain is Legionella pneumophila, is an aerobic, gram-negative, and non-spore-forming bacterium responsible for causing Legionnaires disease, a severe form of pneumonia-lung inflammation [133]. According to the CDC, the infection of at least one out of 10 people by the Legionella bacterium can be fatal. Legionella is transmitted through the inhalation of water droplets, which contain concentrated bacteria [134]. The presence of Legionella is associated with systems containing large amounts of water, such as tanks, air conditioning structures, and fountains [135]. The World Health Organization (WHO) has classified Legionella as the water-borne illness with the greatest impact on human health in the European Union, and many outbreaks are recorded annually around the world.

To date, LOC devices and microfluidics for legionella detection have been presented. For example, Saad et al. [136] specifically designed a Surface Plasmon Resonance imaging (SPRi) titration assay for the detection of Legionella pneumophila cells with the use of an R10C5 aptamer. Aptamers, which are comprised of single-stranded DNA or RNA and function similarly to antibodies, offer superior stability and affordability in comparison to antibodies. The targeted cells were quantified by measuring the unattached cells hybridized onto the surface of the SPRi. For the fabrication of SPRi, Cr- and Au-coated glass slides were utilized. This technique achieved a LOD of 10^4^ cells/mL without the need for amplification. Another study that relies on the fabrication of a biosensor for the detection of Legionella pneumophila sg1 cells in artificial water samples is the work by Laribi et al. [137], who fabricated an electrochemical immunosensor with a gold surface that was modified by the addition of 16-amino-1-hexadecanthiol (16-AHT) for the immobilization of anti-Legionella pneumophila antibodies and, consequently, the capture of the targeted cells via immunological reactions. The targeted cells were quantified using electrochemical measurements and imaging, and the LOD attained was 10 cfu/mL. Another study examines a microchip for the measurement of *L. pneumophila* serogroup 1 in water towers. To identify the targeted cells, a fluorescent antibody was utilized. The method achieved a LOD of 10^4^ cells/mL. The results were more sensitive when the water sample was processed through filtration, where the attained LOD was altered from 10^1^ − 10^3^ cells/mL. The microchip was fabricated using a combination of PDMS and glass and the replica molding technique. In another study, scientists detected *L. pneumophila* cells in water by constructing a surface acoustic wavelength (SAW) immuno-biosensor that uses acoustic waves to penetrate the surface of an elastic material [138,139]. The water sample was enriched with both gram-positive and gram-negative bacteria, as well as anti-legionella antibodies for the specific identification of the targeted cells. The LOD that this device achieved was 2.01 ∗ 10^6^ cfu/mL and the microchamber was fabricated using PDMS [140].

In addition to Legionella, other pathogenic bacteria (such as *E. coli*) can cause severe health problems if they are present in drinking water. To secure water safety, there have been several recent reviews outlining the ongoing research. According to the review paper of Jaywant et al. [141] the development of various microfluidic devices for the detection of bacteria has a significant role in the issue of water safety. *E. coli*, a very important bacterium for the insurance of the water’s quality, can be detected by a plethora of methods that use voltammetry [142], positive Di electrophoresis [143], amperometry [144], integrated electrodes [145], or the Coulter principle [146], via which the changes in electrical resistance are detected and measured. In summary, the vast majority of the materials used for the detection of *E. coli* were mainly glass, PDMS, and silicon, while a paper-based device has also been used. The lowest detection limit of 10 cfu/mL was achieved by the use of integrated electrodes on a modified silicon sensor chip [145].

Except for Legionella and *E. coli*, *S. aureus*, *Salmonella* Typhimurium, and Enterococci are also extremely dangerous bacteria, and methods to detect them accurately in the point of interest have to be developed. Fluorescence [147,148], light scattering [149], and PCR [150] are the most popular detection methods for bacteria detection from real water samples, which originated mainly from sources such as rivers, fields, lakes, seas, or commonly drinking water. The best results were attained by PCR [150] on a PMMA microchip, where the LOD was equal to 6 cfu/mL. For the detection of *E. coli* cells, another research study created a paper-based microfluidic chip. This microchip consisted of three microchannels to detect *E. coli* in different concentrations. One out of the three microchannels was pre-loaded with bovine albumin serum (BSA) to conduct the negative control, while the other two microchannels contained *E. coli* antigens adhered with microbeads. The quantification of the targeted cells was carried out by the Mie Scattering method and screen imaging via an app on a smartphone. The method was extremely sensitive, as it achieved a LOD of just a single cell in a duration of 90 s [149,151]. In another study, a microfluidic chip made of paper was used to identify *E. coli* cells in drinking water. The detection of the targeted cells was aided by bacteriophages engineered genetically. The microchip was manufactured using two injection-molded polycarbonate enclosers. The *E. coli* sample was filtered through a polyvinylidene difluoride (PVDF) membrane. During a period of 5.5 h, the microchip detected 4.1 cfu in 100 mL [152].

Considering the current status of water quality control, we can conclude that a plethora of biosensors is used, yet more devices and approaches are important to be developed because the legislation for safe water intended for human consumption (i.e., in Europe, there is a new Directive (EU) 2020/2184 of the European Parliament and of the Council of 16 December 2020 on the quality of water intended for human consumption) is continuously updated, posing more analyses, including risk analysis, in which microfluidic devices are the perfect candidates for on-site screening tests. In this direction, approaches towards the development of an autonomous analysis micro-device are extremely important [153].

### 3.1. Microfluidics for the Detection of Foodborne Pathogens

Food safety is an area of high interest, and the use of microfluidic devices in this field is highly recommended in order to address foodborne illnesses, which can put at risk public health and have a severe impact on the economy [154]. Food safety concerns both developing and developed nations, such as the United States, where many cases, mostly caused by the pathogenic bacteria of *Salmonella*, *Listeria*, and *Shiga toxin-producing E. coli* (STEC), have been reported from 2009 to 2015. It is estimated that around 800 outbreaks are recorded annually in the United States, with most of them being caused by the consumption of specific food sources (i.e., chicken, pork, and vegetables) [155]. The CDC, a science-based, data-driven organization of the United States, has recorded outbreaks categorized by year of incident, microbes, and contaminated food. From the recorded cases, it appears that from 2020 to 2022, the recent foodborne diseases have been caused by several sub-species of *Salmonella* (i.e., *Salmonella* enteritis, *Salmonella* Typhimurium), *Listeria* monocytogenes and *E. coli* (*E. coli* O157:H7, *E. coli* O121, *E. coli* O103).

Enzyme-linked immunosorbent methods (ELISA), polymerase chain reaction (PCR), and typical culturing methods are the three most frequently employed techniques to detect foodborne pathogens. Even though these procedures are considered highly accurate, in their bench-top mode they are time-consuming and require trained staff and centralized laboratories [156]. We have previously highlighted the advantages of microfluidics and LOCs. It is therefore evident that LOC devices can ensure the quality and safety of food on both an industrial and residential scale. In the following sections, we will discuss recent advances in the use of LOC devices for the detection of *E. coli*, *Salmonella*, *Campylobacter jejuni*, *Listeria* monocytogenes, and *S. aureus*, which are considered the most important pathogenic bacteria in food science.

#### 3.1.1. *Escherichia coli* Detection

*E. coli* is a gram-negative, rob-shaped bacterium found in the intestinal system. *E. coli* exists in numerous forms, most of which are safe. However, some forms, such as the Shiga toxin *E. coli* (STEC), are harmful and can cause foodborne illness. Shiga toxin *E. coli*, whose most important serotype is *E. coli* O157:H7 is related to the uremic syndrome that causes kidney failure and neurological problems. The most common food sources where STEC is present, are raw minced meat, milk, and vegetables that have been contaminated by feces. Strict legislation has been established, setting the concentration of *E. coli* O157:H7 to 0 per 25 gr, in food and water, respectively [157]. 

Biosensors, micro-PCR, ELISA, and Lateral flow assays (LFAs) are the most extensively utilized detection methods at the current time. Immunoassay assays are based on the basic antibody-antigen response, which is used to detect specific microorganisms. Enzyme-linked immunosorbent assay (ELISA), Fluoro-immunoassay (FIA), Chemi-luminescence immunoassay (CLIA), and Radioimmunoassay are the fundamental types of labeled immunoassays (RIA) [158,159]. To measure analyte concentrations, these approaches are linked with MEM biosensors, which are also constructed by employing micro- technologies, and comprise a biological component coupled to a physiochemical transducer that creates signals (optical, magnetic, acoustic, electrochemical, or mechanical in nature [160,161]) proportional to the analyte concentration.

Studies related to the development of LOC devices for *E. coli* detection are presented in this section. Our first example refers to a LOC device used for the detection of *E. coli* O157:H7 in food or clinical samples based on immunoassay reactions. The micro-device integrates a 3D network of microchannels with a micro-biosensor, that measures magnetic changes. The biosensor was coated with Silicon nitride due to its capability of immobilizing antibodies. *E. coli* O157:H7 was detected in quantities of 10^5^ cfu/mL, while non-pathogenic *E. coli* was detected in 10^7^ units/mL [162]. Another MEMS biosensor was created to detect *E. coli* O157:H7 at extremely low concentrations. The biosensor was constructed on a glass substrate and consisted of sensing and focusing regions with distinct functions. The sensing region is responsible for immobilizing bacteria through the specific antibodies that were immobilized. The biosensor demonstrated a lower detection limit (LOD) of 39 cfu/mL within 2 h [163].

A SERS-based microfluidic immunosensor was developed for the detection of *E. coli* O157:H7 in romaine lettuce. For the microchip fabrication, PDMS using soft lithography was employed. For the detection of *E. coli* cells, enrichment and a separate procedure for the targeted cells from the rest of the sample must be conducted. SERS nanoprobes, made of gold particles (AuNPs), and antibodies were used to separate the *E. coli* cells. After 60 min of enrichment, this study detected 0.5 cfu/mL [164]. Another biosensor using gold nanoparticles (AuNPs) was constructed to detect *E. coli* O157:H7 in chicken samples through a color change from blue to red. At high concentrations (5 × 10^8^ cfu/mL), the color change can be observed with the naked eye, while at lower concentrations, a smartphone imaging application was utilized. The microchip consisted of mixing channels, a separation, and a detecting chamber. In the separation chamber, the MNPs-bacteria-PSs structure is separated, while the detection chamber controls the AuNPs’ color change. The microchip was made from PDMS and glass and was modified by surface plasma treatment and 3D printing. The color change occurs when magnetic nanoparticles (MNPs) and polystyrene microspheres (PSs) bind to the target bacterium. This study detected 50 cfu/mL per hour. For improved results, passive micromixers or smaller ones were utilized instead of active micromixers [165].

Prior, we mentioned a SAW immuno-biosensor for the detection of *L. pneumophila* cells in water. Surface acoustic wavelengths are transducers interspersed with the biosensors, aiming to detect enzymes in liquid samples, a task considered to be time-consuming and costly. SAW biosensors can be utilized for the detection of various components (i.e., cells, viruses, proteins, bacteria, and enzymes). In such an example, Tsougeni et al. [166] studied the label-free detection of 1 to 5 cells for multiple bacteria strains (i.e., *Salmonella*, *B. Cereus*, *Listeria*, and *E. coli*) on a LOC using 25 mL milk samples, taking into consideration the active legislation regarding food safety. The proposed LOC device was able to perform all sample handling steps (bacteria capture, bacteria lysis, bacteria DNA amplification, and detection using SAW sensors) on the chip. The microchip was constructed of polymeric materials, and the microchannels were patterned using plasma etching. The analysis duration was 4.5 h, which is almost five times faster than other conventional methods.

Another commonly used method for the detection of pathogenic microorganisms is polymerase chain reaction (PCR), which is an enzymatic assay that helps in the detection and amplification of specific regions of DNA. Recently, PCR was adapted to LOCs in a scheme termed “micro-PCR” [167]. For instance, a polycarbonate microchip was fabricated for the apprehension of *E. coli* O157:H7 cells using the carving technique to design the microchannels. Overall, the microsystem consisted of three microchannels conducting tests simultaneously. In addition, a Quan PLEX platform was used for temperature control, and the amplification of the extracted DNA and fluorescence detection were also utilized. The sample was pre-processed to extract the DNA of *E. coli* O157:H7 and import it into the microchip. As a result, a sample with a low detection limit of 1.2 × 10^−1^ cfu/mL was detected from a large volume of bacteria sample [168].

The printed circuit board (PCB) technology enables the integration of electrodes and, in general, electrical components into microfluidic platforms to achieve a higher degree of integration and simplified fabrication and operation. The manufacturing of PCB-based microfluidic devices is similar to the commonly used fabrication techniques, with the difference that the microchannels are designed in existing layers on the PCB [169]. A representative example of this case is a paper that fabricated a RPA- PCB microfluidic device for the DNA amplification of two *E. coli* fragments. PRA is an isothermal DNA amplification technique that is used instead of PCR. In this work, the microchannels were fabricated using photolithography of dry photosensitive film layers. The main advantage of this technique is associated with energy savings since it only required 0.6 W to work, while the analysis results were comparable with those of a PCR completed using a thermocycler [170] (Figure 11(I)).

It is therefore clear that less complex DNA amplification schemes (i.e., RPA or LAMP, which are isothermal DNA amplification schemes) are used in combination with LOCs.

In another example, estimation of the number of viable target cells is performed with the detection of adenosine triphosphate (ATP). A microfluidic platform was constructed to detect *E. coli* O157:H7 with the combination of the ATP method and the use of immune microspheres for the quantification and, respectively, the detection of bacteria in samples. ATP interacts with luciferin in the presence of a catalyst to produce light energy. The chip consisted of microchannels, which contained aptamers, dendrimers, and micro-barriers in a V-shaped. PDMS and glass were employed for the microchannel’s fabrication, using the soft lithography technique. The results were positive for the detection of *E. coli* bacteria with a LOD of 4.91 × 10 cfu/ μL for a 1 μL of the sample, and the analysis duration was 1.30 h [171]. Another study exploited the P-ELISA method in a paper-based microfluidic analytical device (μPADs) for the detection of various strains of *E. coli* in beef samples. Specifically, hydrophobic barriers made by wax printing in 96 well plates were created to carry out the P-ELISA detection method. For P-ELISA’s performance, two monoclonal antibodies were used, each responsible for a different function (capturing and detection). For the detection, 5 μL of meat sample inoculated with 5 different concentrations of *E. coli* O157:H7 were used. The total operating time was approximately 3 h, and the detection limit was 10^4^ cfu/mL when the conventional ELISA had LOD equal to 10^5^ cfu/mL [92]. A microfluidic platform using PDMS and Soft-Lithography, in combination with a dual RCA, was fabricated to enhance the detection, capture, and targeting of *E. coli* O157:H7 cells in food samples. In this case, orange juice and milk were tested. Dual-RCA results from the merger of s-RCA and c-RCA. RCA is an enhancement technique that modifies surfaces by the addition of aptamers, producing long parts that are composed of DNA sequences and can bind with a multitude of probes, enhancing the detection signals. Dendrimers were also utilized for the modification of the microchannel’s surface as they bond with the aptamers for cell capturing. The dual-RCA can enhance the signals by about 250 times, while the capturing of the targeted cells is increased almost by 3 times. This technique achieved a LOD of 80 cells/mL (Figure 11(II)) [172].

**Figure 11 micromachines-14-00986-f011:**
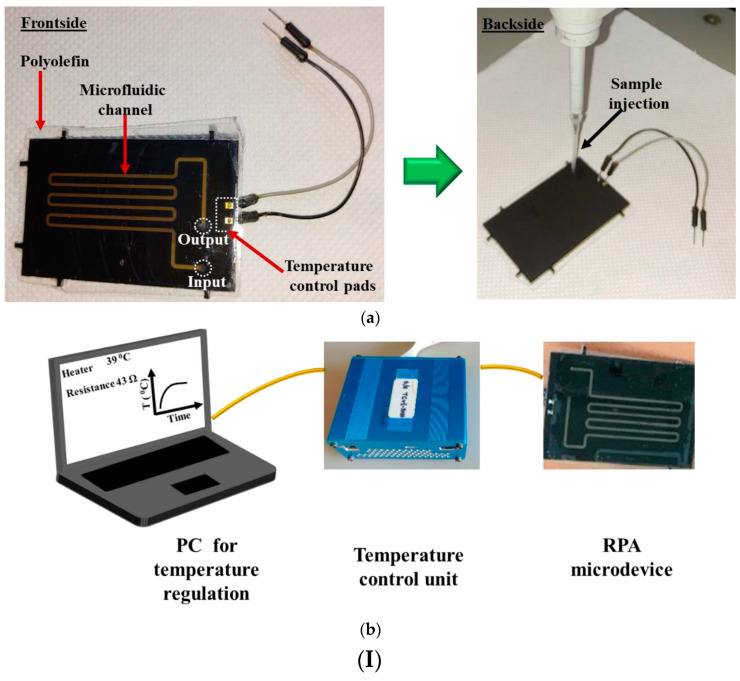
(**I**) (**a**) Image of the LOC on PCB and (**b**) schematic representation of the portable system comprising the LOC, the temperature control unit, and the software for the analysis. Reproduced from Ref. [170] with permission from MDPI Micromachines (CC by 4.0) (**II**) (**A**) Image shows the size of a microchip compared to a coin and (**B**) depicts how target cells are captured by aptamers with immune responses Reproduced from Ref. [172] with permission from Elsevier.

#### 3.1.2. *Salmonella* Strains Detection

*Salmonella* is another bacterium that can cause serious illness, which is Salmonellosis. It is caused by numerous strains of *Salmonella*, a disease that can be transmitted directly or indirectly between humans and animals and results in severe health symptoms and substantial economic losses [173] Salmonellosis may result from contact with an infected carrier or consumption of food or water containing several germs. Raw meat, especially poultry, and eggs, which require special care, are the most common causes of Salmonellosis. To address the detection of three serotypes of *Salmonella* in chicken products, a MEMS biosensor with three distinct microchannels, each for a different serotype, was fabricated. For the microchannel’s fabrication, SU-8 materials were used, while two PDMS plates sealed the microchannel, which was finally placed on a glass slide. Every microchannel consists of a focusing area, responsible for guiding the target cells, and a bacteria-sensing area where immobilized antibody-coated electrode arrays are located. The target cells are guided by polystyrene microbeads. To determine the presence of bacterial cells, the change in electrical resistance is measured before and after their injection. It is worth noticing that no pre-enrichment step was required for cell detection, and finally, the study achieved a LOD of 7 cfu/mL [174].

Another study with a biosensor for the detection of *Salmonella* Typhimurium cells used modified magnetic nanoparticles (MNPs), polystyrene microspheres (PSs), and catalases to separate and enrich the target bacteria from the rest of the sample and transform them into enzymatic bacteria. The bacteria capture was conducted via magnetic forces, while the usage of hydrogen peroxide for washing the PS spheres contributes to an oxygen gap, responsible for cutting off the electrical signal. In this way, the change in the electrical voltage corresponds to the cell’s detection. The microchip was fabricated from PDMS, glass, and acrylonitrile Butadiene Styrene (ABS) using the 3D printing technique. They reported a LOD of 33 cfu/mL for a duration of 2 h [175].

Furthermore, in another work, for the detection of two *Salmonella* serotypes, B and D, in ready-to-eat turkey samples, a biosensor that measured impedance changes was fabricated. The SU-8 biosensor consisted of two detection areas, each having a gold-micro gaped-interdigitated- electrode array. Antibodies of each *Salmonella* serotype were modified with a cross-linker agent to increase the biosensor’s sensitivity to 45–60%. After creating an interface between electrodes and modified antibodies, the sample’s target cells are bound to specific antigens and trapped in the detecting area. The capturing of targeted bacteria changes the impedance, which is measured by an impedance analyzer. This biosensor achieved high sensitivity by separating dead target and *E. coli* cells, and the LOD reached 300 cells/mL in 1 h [176].

A silicone microdevice combined with a nano biosensor that relies on the immune response and the production of fluorescent signals to detect *Salmonella* Typhimurium pathogenic cells in the chicken extract has also been proposed. The employed fabrication techniques were photolithography and plasma treatment. Before the sample is injected, *Salmonella*-antibody-coated magnetic beads are mixed with the sample to detach the target cells. The resulting fluorescent signals are then detected by a fluorometer and are then transmitted to a computer. In summary, the LOD was determined to be 10^3^ cfu/mL in both samples [177].

Immuno-magnetic nanoparticles (MNPs) and fluorescent microspheres (FMSs) were incorporated into a microchip for the detection of *Salmonella* Typhimurium cells in apple juice. For the creation of the microchannel’s mold, SU-8 photoresist and photolithography were initially utilized. The microdevice was fabricated from PDMS and placed on a glass plate. Additionally, surface oxygen plasma was used to improve the adhesion and bonding of the device. A cylindrical filter was employed to disperse the bacteria, and magnetic nanoparticles modified with *Salmonella* monoclonal antibodies were used to separate the target cells from the non-targeted cells (*Listeria* monocytogenes, *E. coli* O157:H7, and Vibrio parahaemolyticus). Bacteria were marked with immunofluorescent microspheres, and the flow of bacteria was recorded using a camera. In a period of two hours, 58 cfu/mL were detected. The study also mentions the relationship between thresholds and fluorescent spots. The optimal *Salmonella* concentrations were between 1.40 × 10^1^ cfu/mL and 1.40 × 10^−6^ cfu/mL, with a cutoff between 10 and 30 cfu/mL [178].

Another method in which fluorescence is used for the quantification of targeted cells is also reported. In particular, a Lab on a Disc device with several microchannels and microchambers was created to detect *Salmonella* enterica cells in egg yolk samples. The microchannels and microchambers were created through the methods of deep-deep reactive ion etching and photolithography. Each of the 24 microchannels on the micro-disc has approximately 300 microchambers. The microchannels directed the sample into the microchambers, where the egg yolk sample was lysed by heat treatment and amplified by PCR with the addition of the invA gene. During the amplification procedure, the number of target cells was quantified using a fluorescent agent. The approach had a significant flaw in that it could not distinguish between viable and nonviable cells. To bind and separate the living cells, they introduced microbeads with anti-*Salmonella* antibodies. This operation occurs prior to the injection of the sample. Their reported detection limit was 5 × 10^4^ cells /mL (Figure 12) [179].

The color change via immunological reaction can operate as an indicator for the presence or absence of a microorganism as well as for its quantification. An immune reaction integrated into a paper-based analytical device (μPAD) for the identification of *Salmonella* typhimurium cells in a culture solution, a sample of bird feces, and a sample of whole milk without pre-enrichment will be presented now. Two paper-based analytical devices with diverse patterns were produced utilizing Whatman No. 4 filter paper and the wax printing method. *Salmonella* antibody-coated magnetic beads bind to the target cells and separate them from the remainder of the sample. Attached to the previous structure (target cell-antibody magnetic beads) is an anti-*Salmonella* antibody with biotin and the -galactosidase enzyme, which reacts with chlorophenol red—D-galactopyranoside (CPRG) to produce a red color associated with the presence of target cells and used to quantify them. After 90 min, 100 cells per milliliter were found in the solution, 10^5^ cfu/gr in the bird feces, and 10^3^ cfu/mL in the whole milk [180].

An innovative method used for the quantification of immune agglutination is Mie scattering, which is based on the refraction of the electromagnetic wave by a homogeneous sphere with a refractive index different from the medium through which it passes. To examine the presence of *Salmonella* Typhimurium cells in a poultry package, a microchip was constructed, consisting of an inlet and two microchannels. The first contained the antibodies, while the second was loaded with Bovine Albumin serum (BSA), which was used as a probe. The sample and the microparticles were vacuum dried to reduce excessive moisture. Immunologically, anti-*Salmonella* antibodies were bonded with the target cells in the microchannels, producing Mie scatter signals. The achieved LOD was 10 cfu/mL for a duration of 10 min. It is worth noticing that the sample was attained without any pre-processing steps from the package to simulate more realistic conditions (no sterile environment) [181]. Two paper-based microchips (one with multiple microchannels and the other with a single microchannel) were fabricated for the detection and quantification of *Salmonella* Typhimurium cells, immunologically and through the Mie scattering method, using a smartphone application that analyzes the captured image via an algorithm. The microchannels were made on cellulose chromatographic paper using a SU-8 negative photoresist, and one target cell was detected in a minute. Due to paper inhomogeneity, wavelength-dependent optical methods may not provide accurate results, unlike the angle-dependent Mie scattering method. In the microchannels, latex microspheres coated with *Salmonella* antibodies are responsible for the reaction and binding with the corresponding cells [182]. 

Lastly, a polymeric microfluidic device was constructed using the injection molding technique, consisting of 8 microchambers for the amplification and detection of *Salmonella* cells in food samples. The DNA amplification is assisted by the Loop-mediated isothermal amplification (LAMP), which is carried out in a single microchamber where the isolation and bacteria detection are also conducted. The 8 microchambers can conduct tests simultaneously, with a LOD of 55 cells/test and a duration of 40 min [183].

#### 3.1.3. *Listeria* Monocytogenes Detection

*Listeria* monocytogenes is a gram-positive bacterium capable of transmission to humans through food sources. The resulting listeriosis is a very serious infection that primarily affects newborns or fetuses during pregnancy [184,185]. *Listeria* monocytogenes can grow on a variety of substrates (decomposed plant matter, animals, moist environments, etc.) and can withstand preventative measures such as refrigeration. Recorded cases of listeriosis in America have been associated with the consumption of raw fruits and vegetables, meat, and raw/unpasteurized dairy products (i.e., cheese, milk, and ice cream) [186,187]. 

A significant amount of research has been devoted to the detection of *L. monocytogenes* in food or bacterial samples. First, a presentation of studies that utilized PCR as a detection method will be presented. More specifically, we will refer to a study that utilized a duplex droplet PCR (ddPCR) for the simultaneous detection of *E. coli* and *L. monocytogenes* cells. The simultaneous function and detection of the two pathogens are conducted by the creation of two different fluorescent probes. The proposed LOC integrates procedures such as droplet generation, amplification, and fluorescence, and it is fabricated with PDMS and a glass plate using the soft lithography technique. It is worth mentioning that, to overcome a serious drawback, the evaporation and crisscrossing of the droplet, a mineral oil-saturated polydimethylsiloxane (OSP) is utilized and added into the microchannels as shown in Figure 13(I). The microdevice was tested in various samples, including drinking water, in which the attained LOD was 10 cfu/mL for both bacteria for a duration of 2 h [188].

Similarly to the aforementioned bacteria strains, the PCR method has also been employed for the detection of *Listeria* monocytogenes cells. An example of this case is the silicone micro-device consisting of four injection nozzles, which was fabricated by the combination of PDMS and SU-8 using the soft lithography technique and oxygen plasma. The first two syringes are used for DNA purification in the micro-chamber, and the third for DNA amplification using real-time PCR and fluorescent dye. PCR runs through thermal cycles, after which wavelengths are generated and filtered to quantify fluorescence intensity, which corresponds to the bacterial concentration. *L. monocytogenes* was detected at 10^4^–10^7^ cells within 45 min [190]. In another study, qPCR was utilized for the detection of both positive (*Listeria* monocytogenes) and negative (*Salmonella* spp.) gram bacteria. A 3D PDMS sponge, as shown in Figure 13(II), was constructed, within which the immunological capture of the target bacteria, their lysis, and, as a result, their amplification via qPCR, occured. The sponge was fabricated from salt crystals, and to be functional, it was treated with oxygen plasma. On the sponge’s surface, ligands capable of binding both types of bacteria are added. In addition, an ApoH protein proved to be more effective than the anti-*Listeria* antibodies. The attained detection limit was between 10^3^ for *Salmonella* spp. and 10^4^ cfu/mL for *L. monocytogenes*, while the effectiveness of the research was determined to be 70% [189].

A microchip combined with an impedance immunosensor is presented for the detection of *L. monocytogenes* bacteria. The materials used for the fabrication of the microchip were PDMS and glass. The capture of the targeted cells is conducted by magnetic nanoparticles modified with antibodies, biotin, and streptavidin. After the injection of the *L. monocytogenes* nanoparticles into the microchip, a change in the impedance is observed, which is measured by an impedance analyzer with the interdigitated electrodes. It is worth noticing that no pre-enrichment step or labeling was required. Furthermore, from this research, 30 nm MNPs were utilized, which have better-capturing efficiency in comparison with larger in-diameter magnetic nanoparticles. The immunosensor was tested for 3 different samples (lettuce, milk, and ground beef), and the achieved LOD was 10^3^ cfu/mL in a duration of 3 h [191].

A paper-based biosensor [192] that detects 198 bp fragments was constructed, which are generated by the application of thermal cycles during the amplification of PCR. The hlyA gene is responsible for activating the enzyme listeriosin O, which is responsible for causing listeriosis [193]. As NaIO_4_ adheres to the biosensor’s surface, it creates a binding site for DNA segments to amplify them. The subsequent addition of NaCNBH3 aids in structure stabilization. The structure reacts with HRP-SA to generate a chemiluminescent (CL) signal that is collected and amplified. Prior to each amplification, a purification process was conducted. The hydrophobic microchip was made of paper using the wax screen printing method, and the attained LOD was 6.3 × 10^−2^ pmol/L.

A microchip consisting of both soft and hard thermoplastic components was created by hot embossing for the detection and capture of *L. monocytogenes* cells. The authors used immunological magnetic beads in a specially designed polymeric microchip. In particular, the surface of the microchip is composed of 3D embossed thermoplastic cylinders coated with soft ferromagnetic nickel. Consequently, once the target cells have bound to the IMNPs, they traverse the specially designed surface and congregate around the pillars due to the strong and alternating magnetic force. The release of cells and a sudden increase in fluorescence intensity result from the deactivation of the magnetic field. The magnetic microchip successfully detected 10 cells per 1 μL, in 1 μL of buffer and beef filtrate [194].

In another effort, PMMA was used for the development of a self-priming compartmentalization (SPC) microchip consisting of three rows and eight columns to visually detect various types of pathogens, including *L. monocytogenes*. In this LOC, microvalves and micropumps for controlling the flow of samples are unnecessary, as the procedure is conducted under negative pressure. The microchip is made from a single plastic layer containing microchannels and holes (inlets and outlets). After degassing, primers are loaded into the microchip, and adhesive tape is used to seal it. For each pathogen type, primers and solution are injected into the microchannel for loop-mediated isothermal amplification (LAMP). *L. monocytogenes* was detected at a concentration of 3.8 × 10^2^ cfu/mL, along with every other pathogen. This SPC microchip and the usage of the LAMP method are considered to be a great solution for the testing of food samples, in which strict regulations have been implemented, such as infant formula, where no pathogens must be detected (Figure 14) [195].

The loop-mediated amplification (LAMP) technique has also been used for the detection of *E. coli* O157:H7, *Salmonella* Typhimurium, Vibrio parahaemolyticus, and *Listeria* monocytogenes on a centrifugal lab on a disc. The lab on a disc was made from PMMA and incorporated many operations, including DNA amplification, extraction, purification, and detection. The results of the DNA amplification were evaluated by a colorimetric test based on the change of color from purple to light blue. The lab on a disc achieved a LOD of 10 cells for every pathogenic bacterium, in a duration of fewer than 65 min [196]. The next work also uses loop-mediated amplification (LAMP), but here it is used in combination with an electrochemical sensor to detect *L. monocytogenes* cells. In particular, a gold concentric-3-electrode was fabricated, consisting of a working, a reference, and a counter electrode. The biosensor and the microchip were fabricated using the 3D printing technique. The LAMP amplification and the specially designed primers could detect 12 serotypes of *L. monocytogenes*. As an electrochemical transducer, the methylene blue redox-active molecule was utilized to specifically distinguish the positive from the negative tests. The attained LOD was 1.25 pg DNA per reaction. The electrochemical biosensor was tested in three food samples: dairy milk, fresh cheese, and smoked salmon [197].

The majority of developed microdevices have low throughputs, which results in longer processing times for a few milliliters of samples to detect the target bacterium. To address this fact, a study processed, separated, and concentrated samples of *Salmonella* Typhimurium, *S. aureus*, and *Listeria* monocytogenes at a high sample flow rate. To do so, they used a microdevice based on magnetophoresis that consists of a magnet and a tube surrounding it. The innovation is that the tube can be replaced with tubes of varying lengths, allowing processing at high flow rates. The microdevice’s tube was manufactured from polyethylene, while the microchannels were created using PDMS and soft lithography. The proposed device was evaluated on two food samples, milk and homogenized cabbage, with the separation and concentration efficiency exceeding 92% and the flow rate above 40%, demonstrating that this device has a better throughput than other devices. Although it is important to note that high flow rates reduce the device’s performance because they create an unsteady flow. To minimize the related vortex, both the tube’s diameter and the microchannel’s shape could be modified [198].

#### 3.1.4. *Staphylococcus aureus* Detection

*Staphylococcus aureus* is a gram-positive, spherical bacterium that inhabits the skin and nasal cavity of humans. Its presence in the human body may not cause symptoms; however, food can be contaminated by human carriers who handle it. The contamination of food with staphylococcus promotes its growth and the production of toxins, which affect the food’s organoleptic qualities, causing external spoilage and a foul odor. *Staphylococcus* is susceptible to heat treatment, whereas the produced toxins are not [199,200]. Numerous studies have been undertaken for the identification of *S. aureus* as well as the assurance of food quality and consumer safety. This section will discuss advances using biosensors and LOCs.

A paper microchip in combination with a biosensor that is based on bimetallic nanoclusters Au/Pt, with peroxidase properties, aiming to identify whole *Staphylococcus aureus* cells, has been proposed. In this case, NCs act as natural enzymes, and reactions with catalysts cause a change in color. Here, TMB and H_2_O_2_ were the catalysts used. The observation of blue color is evidence of the existence of bacteria and is observed by the naked eye but also measured from the change in absorbance at 652 nm. The bacteria quantification is conducted through the paper-based microchip, which is fabricated by Whatman’s No.1 filter paper after CO_2_ laser treatment, forming a Y-shaped micro-device, with a 600 μm Y-shaped microchannel. The color intensity depends on the concentration of bacterial cells (10^2^–10^8^ cfu/mL) and detection limit of 80 cfu/mL was achieved within 5 min [112]. Another study developed a surface-enhanced Raman scattering (SERS) biosensor in conjunction with a microfluidic chip for the detection of two types of *S. aureus*, methicillin-resistant *S. aureus* (MRSA) and methicillin-sensitive *S. aureus* (MSSA), in clinical samples from China and the United States. As far as MRSA is concerned, the detection of this type of *S. aureus* was conducted by the recognition of the mecA gene. The samples were processed with the PCR method and multilocus sequence typing (MLST). The exploited materials were SU-8 photoresist, forming the channel’s mold, PDMS, and glass. The proposed method appeared to be sensitive and could identify the sample’s origin. The biosensor detected 17.400 SERS spectra for a duration of 3.5 h [201].

A microfluidic immunosensor with a nanoporous membrane and specific antibodies to detect pathogen bacteria cells of *E. coli* O157:H7 and *S. aureus* has also been proposed. The immobilization of the target cells is conducted via the self-assembling (3-glycidoxypropyl) trimethoxysilane (GPMS) silane. The impedance change was then recorded for frequencies between 1 and 100 KHz. The device was fabricated by PDMS and attached to aluminum foil. The PDMS layer was treated with oxygen plasma to bond with the aluminum. Sealing of the PDMS to the aluminum was achieved with adhesive epoxy. The immunosensor achieved a LOD of 10^2^ cfu/mL in just 2 h [202]. An aptamer-functionalized graphene oxide (GO) nano biosensor coupled with a hybrid microchip was fabricated, utilizing paper, glass, and PDMS. The proposed micro-device can detect a variety of pathogenic bacteria, including *S. aureus*, and the procedure can be conducted in approximately 10 min. The advantage of this study is that no process of the DNA is necessary. *S. aureus* was detected at concentrations between 5.2 × 10^4^, and 5.4 × 10^5^ cfu/mL in spiked samples [203].

For the detection of the antibiotic resistance mecA gene of *S. aureus*, a self-sufficient lab-on-a-foil system was constructed; the device utilizes recombinase polymerase amplification (RPA) for DNA processing. PDMS was used for the fabrication of this device, implementing techniques such as micro-milling and hot embossing. This lab-on-chip device was constructed to be capable of processing up to 30 samples due to its cartridges. It also achieved the detection of *S. aureus* in less than 10 copies, while the duration of this procedure was less than 20 min [204].

Loop-mediated amplification (LAMP) was employed and integrated into a microchip for the amplification of the mecA and femA genes of *Staphylococcus* in order to detect different types of *Staphylococcus* cells and identify the Methicillin resistance type. More specifically, this micro-device could detect a variety of *Staphylococcus* types, such as *S. aureus*, *S. epidermidis*, *S. haemolyticus*, and *S. hominis*. The material in this case was polycarbonate (PC), which was processed using injection molding. The microchip consisted of 10 microchambers, which were connected to the microchannels. The reported LOD was 20 cfu/reaction for *S. aureus*, *S. epidermidis*, *S. hominis*, and methicillin-resistant *S. aureus* (MRSA) and 200 cfu/reaction for S. haemolyticus in just 70 min [205].

In another example, a PDMS microchip was fabricated on a glass substrate with two intersecting microchannels (an upstream and a downstream microchannel) to conduct PCR on the peanut’s DNA and the hsp gene of *S. aureus*. The microchip was mounted on top of an aluminum block, with a plate heater underneath. (Figure 15). In this way, the microchip was divided into three regions, each with a distinct temperature for carrying out a specific procedure. In the first temperature zone (60 °C), the gene is separated into two single-stranded oligonucleotides. In the second temperature zone (70 °C), the gene extends, and in the third temperature zone (95 °C), the gene denatures. The PCR product is then extracted through the microchip’s outlet. The application of an electrophoresis gel is then used to determine the size of the fragments. Both peanut’s D and *Staphylococcus* target genes were detectable by the microdevice [206].

A microdevice incorporating a plasmon resonance platform, as shown in Figure 15A, for the detection and quantification of *E. coli* and *S. aureus* cells has been presented. As far as *E. coli* is concerned, the platform was able to detect *E. coli* at concentrations between 10^5^–3.2 × 10^7^ cfu/mL in a solution containing phosphate-buffered saline (PBS) and peritoneal dialysis fluid (PD), whereas *S. aureus* was contained in PBS. The microchip was made of PMMA and consisted of a single microchannel, a sample injection inlet, and an outlet. The microchip was also gold-coated and modified with specific antigens to capture *E. coli* and *S. aureus* cells. Detection and quantification were performed by bright field and fluorescence imaging [207,208].

A self-made microfluidic chip was fabricated for the detection of *S. aureus* via labeling and the use of immune microspheres with 50–90 nm diameter for surface modification. After a 4-min procedure (reaction), *S. aureus* was detected at a LOD of 1.5 × 10^1^ cfu/ μL. Practically, the procedure of detecting the target cells is based on the binding of the target cells with specific antibodies, that bind respectively with antigens. Then the antigens bind with the anti-*S. aureus* antibody (FITC). For the creation of the mold spin-coating SU8-3025 photoresist was used and processed by silane treatment. After 5 min, PDMS is poured and baked at 80 °C for 10 min. For 2 μL/min flow rate, the fluorescence intensity reaches its maximum functionality, while a 5 μL/min flow rate and a 4 min duration are the optimal conditions for the optimal injection and reaction. At this flow rate, the target bacteria can be seen from the microsphere’s surface [209].

#### 3.1.5. *Campylobacter* spp. Detection

*Campylobacter* is a pathogenic, gram-negative, non-spore-forming bacterium responsible for many outbreaks of foodborne illness. Morphologically, it is mainly found in an S-robbed shape and has a length of 0.5–5 μm. *Campylobacter* is sensitive to atmospheric oxygen and can be killed by heating. Nevertheless, it shows the ability to persist in contaminated food, one of the reasons being the development of biofilms, which are strong structures with high resilience. *Campylobacter* is present in animals, through which it is transferred to humans or animal-derived products. The products that most often contain campylobacter are raw meat, with the most basic form being chicken, and raw/unpasteurized milk. However, campylobacter can contaminate water or even ice [210,211,212].

In the context of combating the antibiotic resistance crisis, Ma et al. [213] developed a rapid detection method for antimicrobial resistance surveillance in agri-food systems. More specifically, a polymer-based microfluidic device was developed for the identification of *Campylobacter* spp. and AST. 100% specificity was achieved in detecting multiple *Campylobacter* species in artificially contaminated milk and poultry meat. On-chip AST determined *Campylobacter*’s susceptibilities to multiple clinically significant antibiotics with a high degree of concordance (91–100%). The lab-on-a-chip device completed on-chip identification and AST in 24 h, whereas traditional methods require days. The developed method decreased analysis time, streamlined food sample preparation and chip operation, and utilized up to 1000 times fewer reagents than standard reference methods, making it competitive for rapid screening and surveillance studies in the food industry. This microfluidic device could be used for food safety management and clinical diagnostics in areas with limited resources.

Building on the capabilities of the dual-sample-on-a-chip LAMP assay, J. Jin et al. [214] created a microfluidic chip with an integrated loop-mediated isothermal amplification (LAMP) technique to detect ten waterborne pathogenic bacteria, including *Campylobacter jejuni*. The method simultaneously completed 22 genetic analyses in an automated format, with a detection limit ranging from 7.9 × 10^−3^ to 9.54 × 10^−1^ pg of genomic DNA of pure bacteria per reaction, and the LAMP reaction was completed in just 35 min. In the analysis of coastal water samples, the clinical sensitivity and specificity were 93.1% and 98.0%, respectively. The established dual-sample on-chip LAMP assay provides a dependable and efficient multiple-pathogen analysis of waterborne bacterial pathogens and is appropriate for on-site detection and routine monitoring of waterborne bacteria in aquatic environments. A droplet PCR method for the identification of pathogen DNA biomarkers utilizing polystyrene micro-beads with fluorescent color coding has also been proposed. More specifically, using a commercial bead set, the authors were able to encode several singleplex droplet PCRs. It is highlighted that this technique provides more scalability than the limited quantity provided by fluorescent detection probes like TaqMan probes. The approach was verified for three distinct bead sets coupled with target-specific capture oligonucleotides to detect the hybridization of three pathogens infecting poultry: avian influenza, infectious laryngotracheitis virus, and *Campylobacter jejuni*. The target DNA was amplified using fluorescently labeled primers and monodisperse picolitre droplets. The hybridization technique recognized the target DNA of all three species with high specificity from samples containing an average of one DNA template molecule per droplet. It is also worth mentioning that, with the scalability enabled by the color-coded detection beads, the droplet PCR assay detection panel could simultaneously detect many targets [215].

Using a microfluidic device for the continuous separation of bacteria from food samples, a recent study proposes a novel method for pathogen detection that builds on previous research. A microfluidic “LabDisk” capable of parallel, PCR detection of up to six distinct food-borne pathogens. It is composed of a microfluidic network for integrating positive controls, no-template controls (NTCs), and standards (STDs) into a centrifugal microfluidic PCR cartridge. In qualitative mode, each cartridge may test two DNA samples for the presence of six food pathogens, including PCs and NTCs, which are *Listeria monocytogenes*, *Salmonella typhimurium*, EHEC, *S. aureus*, *Citrobacter freundii*, and *Campylobacter jejuni*. This was validated for 10 pg, 1 pg, and 0.1 pg per pathogen DNA quantities. In quantitative mode, one DNA sample per cartridge can be quantitatively analyzed for the presence of two pathogens using standard curves that have been pre-made and created on a disk. 50 pg and 500 pg samples of *L. monocytogenes* genomic DNA were measured to contain 83 ± 17 pg and 540 ± 116 pg DNA, respectively, while 50 pg and 500 pg samples of S. Typhimurium DNA contained 48 ± 4 pg and 643 ± 211 pg DNA, respectively. It is shown that in both operating modes, microfluidic liquid routing was accomplished by spinning the cartridge on a low-cost centrifugal test apparatus [216].

Lastly, Pires & Dong [217] detected waterborne pathogens such as *E. coli* O157:H7, *Campylobacter jejuni*, and adenovirus to ensure the safety of water resources. In this study, a poly (methylmethacrylate) (PMMA) microfluidic biosensor was developed and integrated with an array of organic blend heterojunction photodiodes (OPDs) for chemiluminescent detection of the target pathogens. The PMMA chip allowed for the capture of pathogens, which were then detected by poly(2,7-carbazole)/fullerene OPDs with a sensitivity greater than 0.20 A/W at 425 nm. The limits of chemiluminescent detection for *E. coli*, *C. jejuni*, and adenovirus were determined to be 5 × 10^5^ cells/mL, 1 × 10^5^ cells/mL, and 1 × 10^8^ mg/mL, respectively. In less than 35 min, the integrated biosensor enabled the simultaneous analysis of all three analytes. As illustrated by the recovery tests, the study also demonstrated the capability of the integrated biosensor to detect bacteria in actual water samples. The findings indicate that the PMMA-OPD biosensor can be used to detect waterborne pathogens rapidly and sensitively in environmental samples.

In Table 3, we summarize the most relevant and representative LOC efforts existing in the literature for the detection of the most important pathogenic bacteria that can be found in food products.

## 4. Conclusions and Discussion

### 4.1. Research Outlook of Microfluidics and LOCs

Microfluidic chips and LOCs are miniaturized devices that can exploit phenomena such as the capillary flow of fluids in conjunction with technologies like PCR and can offer attractive solutions for the on-site detection of several pollutants. They can also serve as an important tool for quality control in food and water, as well as in the medical industry, where they are primarily applied. From what was presented in the previous sections, it is clear that a substantial amount of research has been undertaken in relation to food quality and pathogenic microorganisms. Since *E. coli*, *Salmonella*, and *Listeria* are the most common bacteria discovered in food matrices, these are the microbes that have received the most attention in the scientific community. Regarding the fabrication techniques to produce microchips, thermoplastics are the most common materials used in microfluidics since there is a demand for cheap and disposable microfluidic devices. Microfluidics and more complex LOC devices are usually made of thermoplastics, because of the existing mass-scale production methods existing (injection molding) and their unique properties (i.e., biocompatibility, rigidity, transparency, resistance to several solvents, etc.), which enable coupling with established detection methods. Such approaches can provide LOC devices with low detection limit capable for real life applications.

There is also great interest in the paper-based chips due to their low cost and user friendliness. Paper-based devices are compact and lightweight, meaning they are ideal for use in point-of-care applications and also an eco-friendly option since paper is a biodegradable and environmentally friendly material. However, the sensitivity of μPADs is usually limited; they cannot meet the regulatory requirements in food microbiology, and this is one of the main reasons that most efforts have not yet been commercialized.

For the detection of pathogenic microbes present in food and/or water samples, a wide range of detection techniques are available. In this review paper, we presented recent studies, each of which utilized or combined different detection techniques for the identification of various bacteria. In conclusion, the detection methods that are implemented into LOCs can be categorized into four main categories, Biosensors, Nucleic acid-based assays, immunoassays, and spectroscopic techniques. Biosensors are usually coupled with microfluidic devices, and they can be used to detect changes, such as magnetic, voltammetry, impedance, color, or resistance. In addition to biosensors, there are techniques based on nucleic acids that are used to detect pathogenic microorganisms. These methods rely on the detection of pathogen-specific DNA or RNA sequences. qPCR, for example, is a commonly used method for the identification of bacteria and is based on the use of fluorescent dyes or probes to measure the quantity of DNA or RNA amplified during the PCR reaction. These methods are currently very popular because they offer a very low limit of detection. Immunoassays employ specific antibodies capable of binding to the pathogenic microorganisms of interest, and the detection is completed using a label. Immunoassays are really fast and easy to perform, but again, in most cases, they exhibit limited sensitivity and can be mainly used as screening, rapid tests in food microbiology. Finally, the spectroscopic techniques such as Raman spectroscopy and surface-enhanced Raman spectroscopy (SERS) can be incorporated into LOC devices to create highly accurate, portable, and miniaturized analytical instruments for various medical, environmental, and chemical applications. Overall, all these methods have been successfully implemented in LOCs for research purposes, usually under ideal conditions, and few of them have passed to the next level of commercial exploitation. Therefore, there is a need for research and development on existing methods rather than proposing new ones, utilizing them on real samples and in industrially relevant settings, in order to accelerate the industry’s uptake of already developed concepts. In this direction, the technological advancements that have significantly simplified the detection protocols (i.e., the LAMP method does not require DNA purification) and reduced the number of pretreatment steps are expected to provide more “success stories” of microfluidic based and LOC approaches being commercialized during the next few years.

Other state-of-the-art applications of microfluidics and LOCs in food science also include, but are not limited to, microfluidics to test the antibacterial action of emulsions or antibacterial agents, microfluidics to synthesize antibacterial composites and particles, as well as microfluidics used in nanoencapsulation of functional substances.

### 4.2. Industrial Outlook of Microfluidics and LOCs

As far as the industry is concerned, the market for microfluidic chips is projected to reach $7.1 billion by 2025, indicating strong development potential. The fast expansion of the point-of-care testing industry and the rising need for downsizing and automation of laboratory procedures are driving market growth. Geographically, the microfluidic chip market is dominated by North America and Europe, with the United States being the largest contributor. Asia-Pacific, led by China and Japan, is also rising as a significant area. These areas have a substantial market presence due to factors such as technological breakthroughs, favorable government regulations, and the existence of key industry players. Infrastructure, strong investment in research and development (R&D), the presence of key industrial players, and favorable government assistance are all characteristics of a prosperous economy.

The United States has emerged as the industry leader for microfluidic chips, holding a large market share. This is the result of several variables, including a well-established healthcare infrastructure, substantial investment in research and development (R&D), the presence of large industry players, and supportive government policies. The U.S. healthcare system is robust, with considerable demand for innovative medical technology and products. This, in turn, increases the need for microfluidic chips, which are indispensable for medical diagnosis and treatment. The United States offers a favorable climate for R&D, with considerable government and commercial investment. This investment has resulted in innovation in the microfluidic chip market as companies strive to produce new and innovative products.

While it has many benefits, it is crucial to highlight that LOC devices appear to have some drawbacks; despite being a very large market, LOC devices are not yet mass-produced, resulting in higher prices. There is a link between the quantity of created devices and their price. Moreover, the competent authorities (i.e., the FDA) have not yet allowed their extensive usage in quality control/safety. Another significant disadvantage is that in most cases up to now, LOC devices have been predominantly evaluated under optimal conditions, creating doubt regarding the precision and dependability of the results under sub-optimal situations. Therefore, more examination and testing are necessary to guarantee their full functionality.

## Figures and Tables

**Figure 1 micromachines-14-00986-f001:**
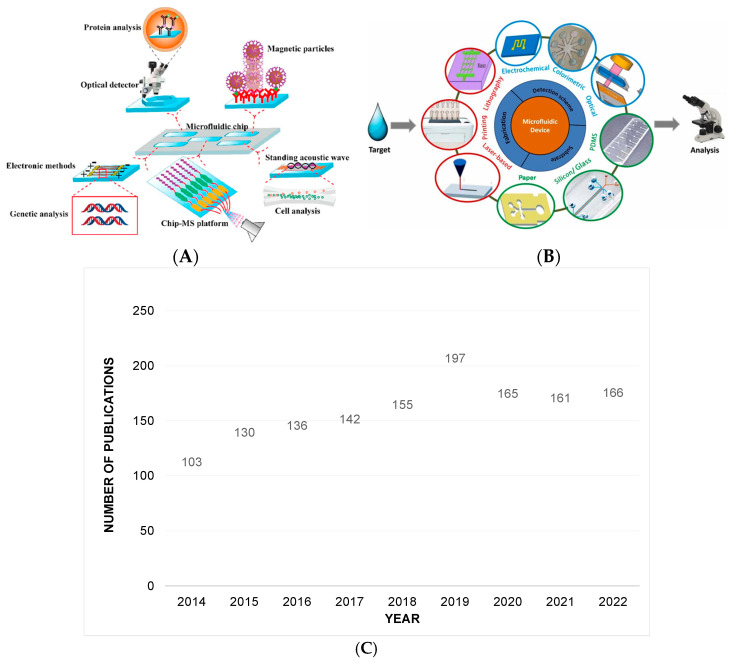
(**A**) This figure depicts the basic parts of a microfluidic device and some key targets analytes (i.e., proteins, cells, DNA, etc.). Reproduced with permission from Ref. [14] Elsevier in Analytical Chemistry. (**B**) This figure summarizes all the fundamental LOC device detection methods, materials, and fabrication techniques. The most commonly used detection techniques are electrochemical, optical, and colorimetric, and the most commonly employed materials are paper, PDMS, and silicon/glass. Lithography, printing, and laser-based techniques are widely used methods for microfluidic fabrication. Reproduced with permission from Ref. [15] Elsevier Sensors and Actuators A: Physical. (**C**) The number of papers published during the period 2014–2022 in the thematic area: Microfluidics/LOCS for bacteria detection in Food.

**Figure 2 micromachines-14-00986-f002:**
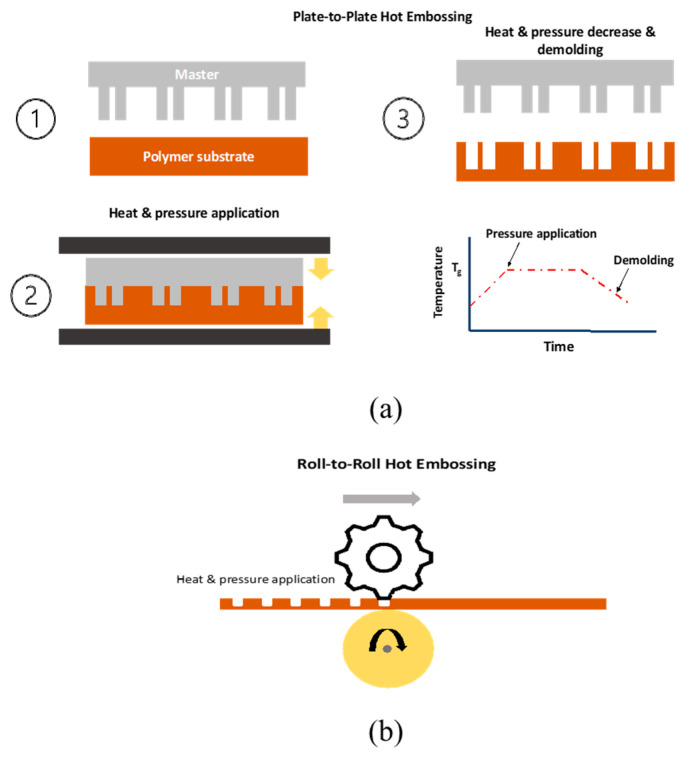
(**a**) Schematic representation of the main steps during hot embossing. ①–③: The polymer substrate is placed between the two heated plates in which also pressure can be applied (i.e., heated press). The temperature is set to or above the glass transition of the polymer, then pressure is applied, and the pattern is transferred to the softened polymer. Eventually, the temperature is decreased, and the pressure is released. (**b**) Roll-to-Roll hot embossing is a continuous process and preheating the substrate is not necessary. The upper roll contains the master that is heated, constant pressure is applied, and the pattern is transferred to the substrate.

**Figure 3 micromachines-14-00986-f003:**
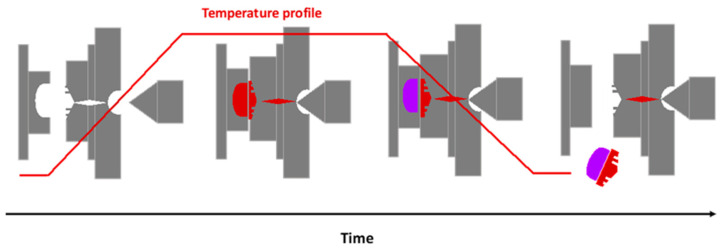
Schematic illustration of the injection molding process. The temperature profile versus time is also provided. The thermoplastic material is fed into a heated barrel and injected into a mold cavity where it is left to cool down and harden. Eventually, the product or the microfluidic device is fabricated.

**Figure 4 micromachines-14-00986-f004:**
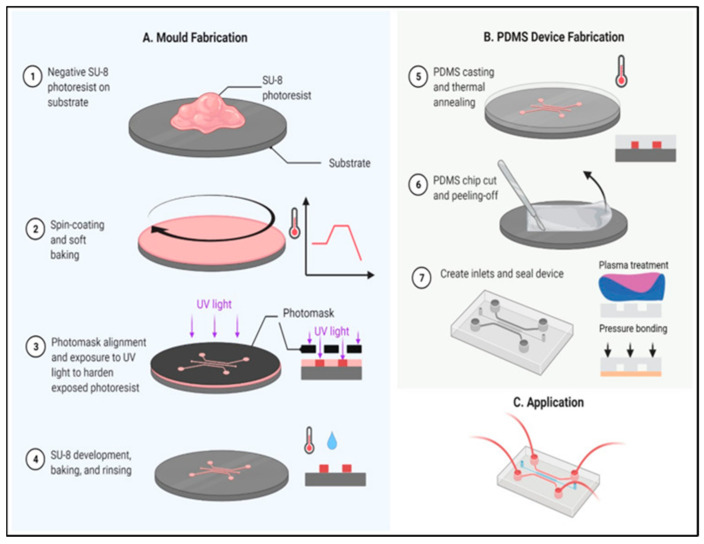
(**Left**) mold fabrication process using SU-8. (**Right**) PDMS casting on the SU-8 mold for the development of a microfluidic device and subsequent bonding after plasma treatment to seal the device. Reproduced with permission from [25] (CC by 4.0).

**Figure 5 micromachines-14-00986-f005:**
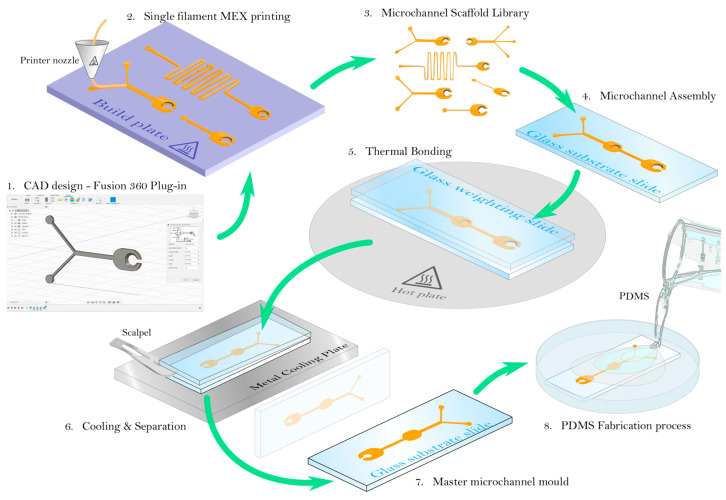
CAD design and 3D printing of the microfluidic channel, which is used as a mold to produce PDMS microfluidic chips. Reproduced with permission from [67] (© 2023 Felton et al.), (CC by 4.0).

**Figure 6 micromachines-14-00986-f006:**
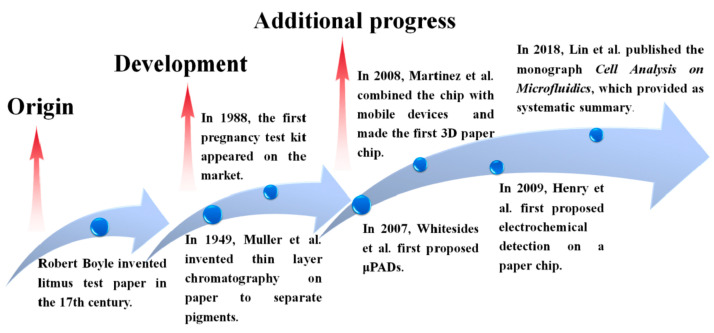
Milestones in the development of microfluidic paper-based chips. Reproduced with permission from [87].

**Figure 7 micromachines-14-00986-f007:**
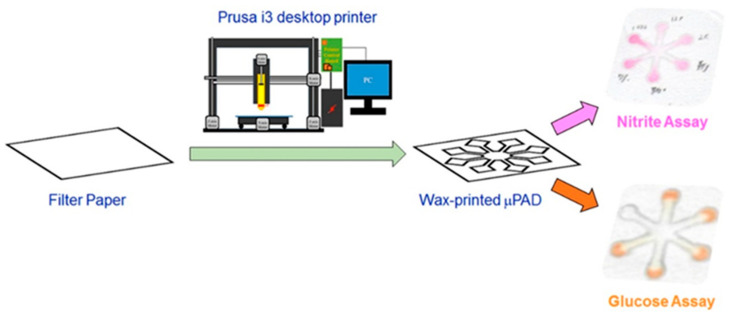
Fabrication of μPADs with wax printing: Paraffin wax is printed on filter paper, then it cools and becomes solid while forming the desired microfluidic patterns. Reproduced with permission from Ref. [93].

**Figure 8 micromachines-14-00986-f008:**
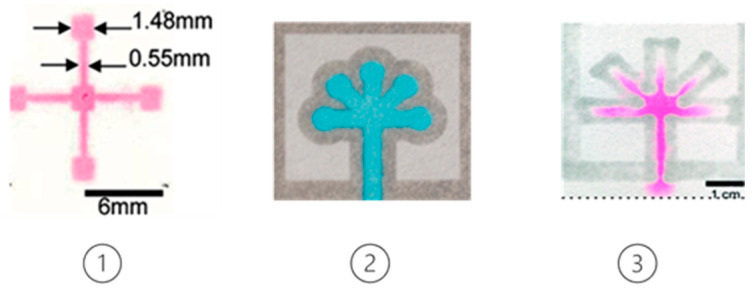
① Using inkjet printing of poly(styrene) with toluene and colored ink, to visualize the paper’s structure. Reproduced with permission from Ref. [85] 2008 American Chemical Society. ② Micro-PADs fabricated via printing a pattern on chromatography paper with an ink-jet printer. Reproduced with permission from Ref. [82] PMC Open Access Subset. ③ Hydrophilic channels can be created on paper by printing hydrophobic PDMS, followed by introducing a 10-μL sample into the channels. Reproduced with permission from Ref. [103] © 2023 Copyright American Chemical Society.

**Figure 9 micromachines-14-00986-f009:**
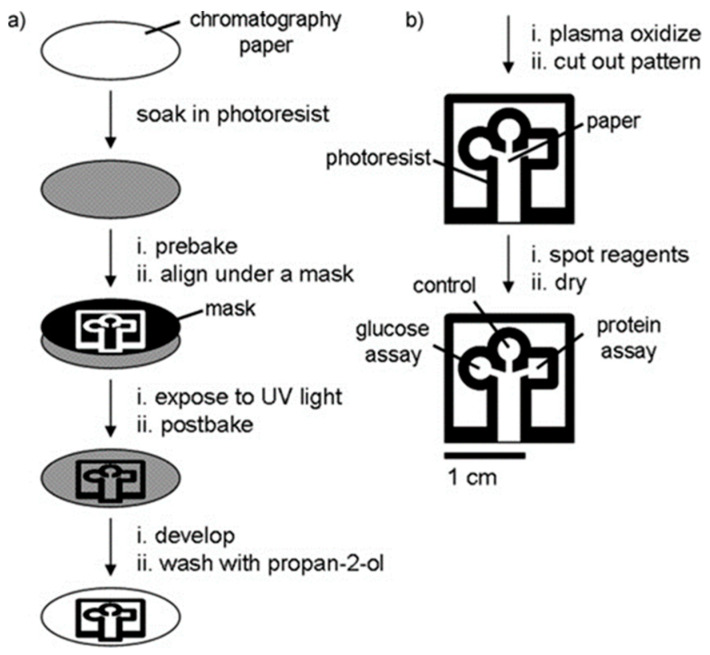
Optical lithography is a high-quality technique with many applications. The basic process of photolithography on paper-based microfluidics involves impregnating an entire sheet of paper with a negative photoresist, then exposing it to UV light through a photomask to crosslink the photoresist in the desired pattern and developing the paper in solvent to remove any unexposed resist. (**a**) Optical lithography on paper for the fabrication of a paper based device and (**b**) Plasma processing, cut out of the device and final modification steps on the paper based device. Reproduced with permission from Ref. [31].

**Figure 10 micromachines-14-00986-f010:**
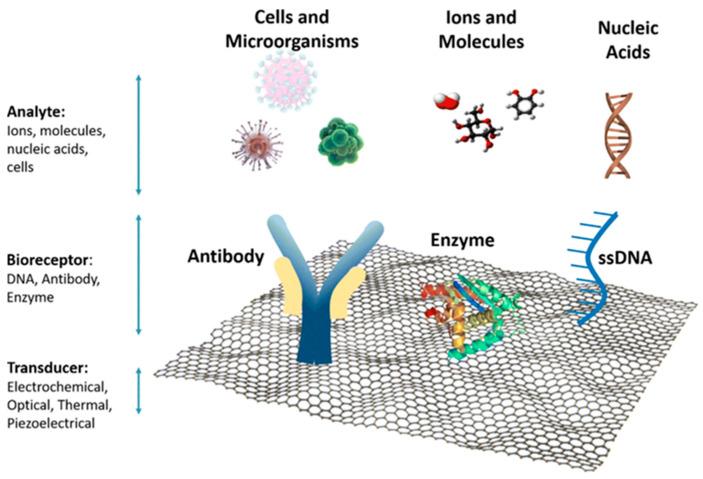
This figure depicts the three components that comprise a biosensor. First, the analyzer identifies biological elements (i.e., cells), the bioreceptor, which is responsible for binding target analytes, and the transducer, which converts current energy into other forms. This biosensor is positioned on top of a graphite substrate. Although different stable, transparent, stretchable, biocompatible, and transportable materials can be used in place of graphite as a biosensor’s substrate. Reproduced with permission from Ref. [132] (CC by 4.0).

**Figure 12 micromachines-14-00986-f012:**
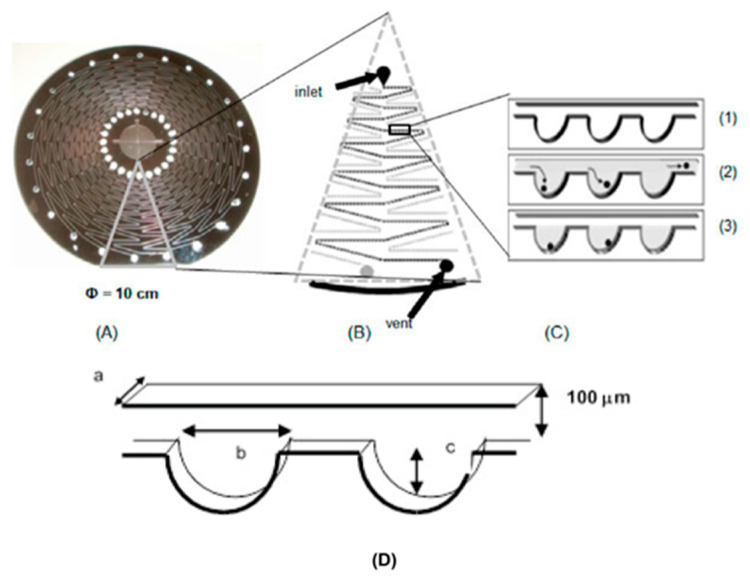
(**A**) Schematic illustration of the microfluidic disc, consisting of 24 inlets and outlets respectively, in order to conduct tests simultaneously. The inlet allows the fluid to enter and cross the microchannels, which form zig-zag structures. (**B**) The second image focuses on a part of the microchip, where the microchannels are crisscrossing, forming a larger zigzag. (**C**) The third image focuses on a microchannel’s part, that consists of multiple and in-a-row microchambers, in which targeted cells are trapped by the centrifugal force. (**D**) The dimensions of a microchamber, into which there are shaped cavities as well as the width of the microchannel, which corresponds to 100 μm are depicted. Reproduced from Ref. [179] with permission from MDPI Micromachines (CC by 4.0).

**Figure 13 micromachines-14-00986-f013:**
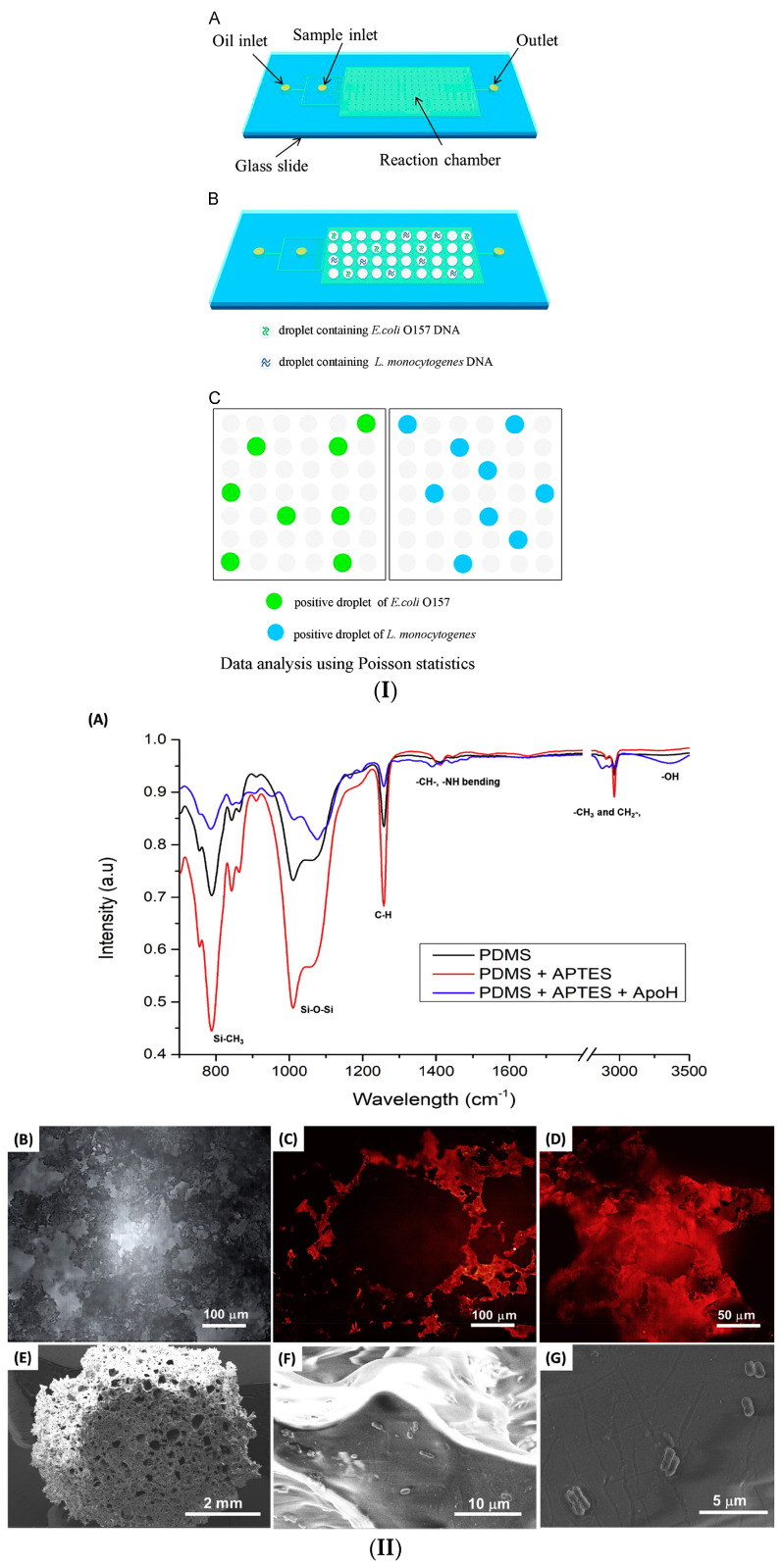
(**I**) (**A**) A schematic representation of a microfluid platform placed upon a glass substrate is depicted. The microdevice is composed of two inlets, one for the sample and the other for the mineral-saturated polydimethylsiloxane oil. The addition of the oil has the purpose to prevent the evaporation and crisscrossing of the droplet. Furthermore, in the center of the microdevice, there are the reaction chambers where fluorescent probes, specially designed to bind with *L. monocytogenes* or *E. coli* are contained. (**B**) An illustration of the microfluidic chamber, where the two bacteria have been trapped and detected. (**C**)The detected bacteria have reacted with the fluorescent probes, producing color. The green color corresponds to the positive test for *E. coli* identification, while the blue corresponds to *L. monocytogenes*. Reproduced from Ref. [188] with permission from Elsevier B.V. Biosensors and Bioelectronics. (**II**) (**A**) From the above diagram we conclude that the PDMS + APTES + ApoH structure has a better intensity in comparison with the PDMS + APTES and the simple PDMS. In addition, we notice that from 600 to 3000 wavelength there is a difference in intensity between the 3 structures, while from 3000 and up the difference is insignificant. (**B**) An image of the sponge prior to the fluorescence (**C**) Picture of the fluorescent PDMS sponge (**D**) A cross-sectional view of the PDMS sponge after bacterial capturing (**E**) PDMS sponge and its natural cavities (**F**) Targeted bacteria trapped in the cavities of the sponge, that works correspondingly to the microchambers (**G**) An up-close image of the bacteria on the sponge’s surface. Reproduced with permission from Ref. [189] with permission from Elsevier B.V. Food control.

**Figure 14 micromachines-14-00986-f014:**
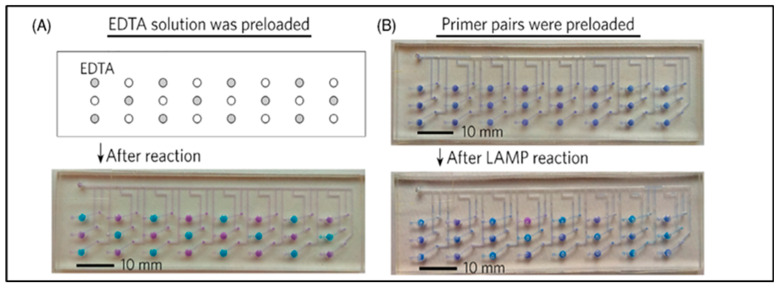
(**A**) In figure A the sites where EDTA has been loaded are depicted, resulting in a noticeable change in the color after the reaction has occurred. The color change is depending on whether there is a negative or positive control. (**B**) In the second figure, the primers were pre-loaded on the microchip, for the capturing of the targeted cells, which causes a color change. In both images, the distance between the outputs is indicated and corresponds to 10 μm. Reproduced from Ref. [195] with permission from the Society of Chemical Industry.

**Figure 15 micromachines-14-00986-f015:**
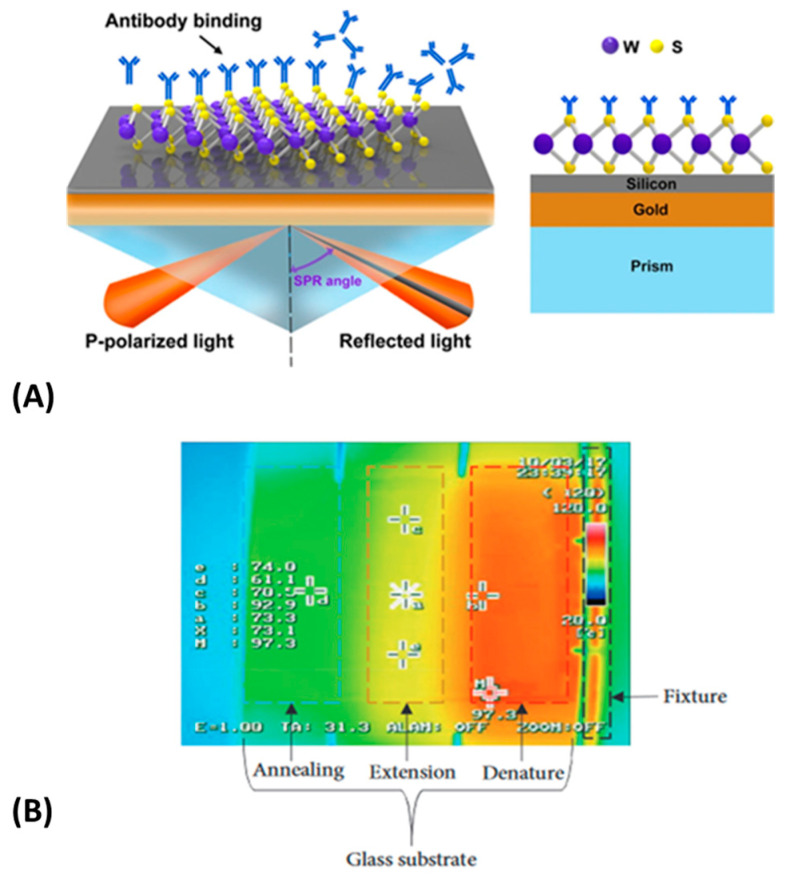
(**A**) In this figure, a microchannel is depicted in which antibodies bind to WS2. The substrate of the microchannel is composed of silicon, and beneath it is a gold film. Additionally, polarized light is positioned in the microchannel’s center. The light source emits polarized incident light, while the detector collects the reflected light, whose intensity is diminished due to the resonance angle (SPR). Surface plasmon resonance permits optical investigation of the molecular interaction between a moving molecule and a fixed molecule on a substrate. Reproduced with permission from Ref. [207] CC by 4.0). (**B**) An illustration of the heater plate is shown in which the plate is divided into three sections, based on the conducted procedure and the necessary temperature for the PCR amplification. Reproduced with permission from [206]. Copyright © 2023 Shou-Yu Ma et al.

**Table 1 micromachines-14-00986-t001:** Materials and fabrication methods for polymer-based LOCs. The advantages and disadvantages of each method are also provided.

Fabrication Method	Material	Advantages	Disadvantages	Reference
Hot Embossing	PMMA COP	Resolution of several tens to hundreds of micrometers Different modes of heating (e.g., Far Infrared, ultrasonic)/ Isothermal/non-isothermal heating 2 types of operation (plate-to-plate, roll-to-roll) No cleanroom requirement Good repeatability	Difficulty in demolding Significant residual thermal stress Master cost	[35,36,39,40,41]
Injection molding	COC PMMA	Good repeatability Fast method Mass production High quality Process of 3D microfluidic chips No cleanroom requirement	Incomplete filling Not always homogenous melting. Mold cost	[46,47,48]
Casting (Soft lithography)	PDMS	Low cost Easy setup Large surface area patterning No cleanroom requirement	Need for fabrication of a master. Distortion of elastomeric materials.	[51,52]
Micromachining techniques	COP PMMA PS/PC PET PDMS Biodegradable polymers	Versatile Plethora of applications Flexible No cleanroom requirement	High heat-affected zone (HAZ) can affect the properties of the material. Limited resolution	[53,54,55,56,57,58]
3D printing	Polylactic acid (PLA) Acrylonitrile butadiene styrene (ABS)	No cleanroom requirement Low cost High accessibility Rapid production Multiphase printing	Limited materials Post-processing Limited resolution	[62,63,64,65]
Lithography techniques	SU-8	High resolution Flexible method Relative high cost of photoresists	Need for cleanroom	[68,69,70]
Plasma processing	PMMA	Functionality incorporation and patterning Fast method High resolution	Expensive equipment System dependency Limitations on sample size	[71,72,73,74,75,76,77,78]

**Table 2 micromachines-14-00986-t002:** Materials and methods used for the fabrication of paper-based LOCs.

Fabrication Method	Material	Advantages	Disadvantages	Reference
Wax printing	Wax	Fast Low cost Mass production Simplicity	Limited resolution	[91,92]
Inkjet printing	PS/hydrophobic sol-gel (MTMS)/Silicon/Alkyl ketene dimer/Polyacrylate/TiO_2_ nanoparticles in polyurethane	Low cost Mass production Easy operation Direct printing	Limited resolution	[94,95,96,101,102]
Photolithography	Poly (o-nitrobenzyl methacrylate) (PoNBMA)/ Octadecyl trichlorosilane (OTS)/SU-8/PMMA/cyclized poly (isoprene) derivative photoresist/ Propanediol methyl ether acetate (PGMEA)/Polyurethane acrylate	High resolution Good stability Flexibility	High cost of photoresists Equipment High number of steps	[31,102,105,106,107,108]
Screen printing	Conductive carbon-based ink Biodegradable polymers	Versatility Compatible with a variety of inks	High number of steps Time-consuming High cost	[110]
Laser processing	Directly on paper	Versatility High accuracy	Paper damaging Expensive equipment	[111,112]
Plasma processing	Alkylketene dimer	Reproducibility Flexibility Good resolution	Expensive equipment	[114]
3D printing/Lamination methods	Resin	Low cost Fabrication of complex structures	Limited resolution Not suitable for mass production	[109,115,116,118,119,120,121]

**Table 3 micromachines-14-00986-t003:** Summary of the reviewed studies, categorized by pathogen microorganism reported together with the material used, the fabrication method and limit of detection.

Detection Method	Material	Sample	LOD	Fabrication Method	Reference
** *Escherichia coli* **					
Biosensor magnetical changes	SU-8 photoresist	Food sample	10^5^ cfu/mL	3D stereolithographic technique	[162]
MEMs biosensor	SU-8/glass	Bacteria samples	39 cfu/mL	Photolithography/surface micromachining	[163]
SERS immunosensor	PDMS	Romain lettuce	0.5 cfu/mL	Photolithography/Soft lithography	[164]
PDMS coated Biosensor Color	PDMS/glass	Chicken sample	50 cfu/mL	3D printing/Surface plasma treatment	[165]
SAW biosensor	Polymeric materials	Milk sample	1–5 cells	Plasma treatment	[166]
micro-PCR	Polycarbonate	Bacterial sample	1.2 × 10^−1^ cfu/mL	Carving technique	[168]
PRA, isothermal amplification	PCB photosensitive dry film	Bacterial sample	DNA amplification achieved		[170]
ATP assay	PDMS/Glass	Bacterial sample	4.91 × 10 cfu/μl	Soft lithography	[171]
P-ELISA	Paper	Beef sample	10^5^ cfu/mL	Wax printing	[92]
Dual RCA	PDMS	Orange juice/milk	80 cells/mL	Soft lithography	[172]
***Salmonella* spp.**					
MEMs impedance biosensor	PDMS SU8	poultry products	7 cfu/mL	Surface micromachining Oxygen plasma	[174]
Impedance biosensor	SU-8 PDMS glass	ready-to-eat turkey sample	300 cells/mL	Surface micromachining Oxygen plasma	[176]
Immuno-biosensor Fluorescence	Silicone PDMS/glass	Chicken extract Borate buffer	10^3^ cfu/mL	Photolithography Plasma treatment	[177]
Immune magnetic nanoparticles (MNPs) Fluorescent microspheres (FMSs)	PDMS glass	Apple juice	58 cfu/mL	SU-8 photolithography Oxygen plasma treatment	[178]
PCR fluorescent factor	Silicon wafer glass	egg yolk	5 × 10^4^ cells/mL	ion etching photolithography	[179]
Immunological reactions color change	paper	bird feces whole milk	100 cells/mL 10^3^ cfu/mL	Wax printing	[180]
Mie scattering	acrylonitrile-butadiene-styrene (ABS)/polycarbonate	poultry package	10 cfu/mL	Micro-milling	[181]
Screen imaging Mie scattering	cellulose chromatographic paper	Bacterial sample	1 cell	SU-8 negative photoresist	[182]
Biosensor	PDMS Glass acrylonitrile Butadiene Styrene (ABS)	Bacteria sample (*E. coli*, *Salmonella*, *L. monocytogenes*)	33 cfu/mL	3D printing	[175]
LAMP amplification	polymeric	Food sample	55 cells/test	Injection molding	[183]
** *Listeria monocytogenes* **					
ddPCR	PDMS glass	Bacterial sample Drinking water	10 cfu/mL	Soft lithography	[188]
PCR fluorescent dye	PDMS SU-8	Bacterial sample	10^4^ cells	Soft lithography Oxygen plasma	[190]
qPCR	PDMS sponge Salt crystals	Bacterial culture	10^4^ cfu/mL	Oxygen plasma	[189]
impedance immunosensor, modified magnetic nanoparticles	PDMS glass	Lettuce Milk Ground beef	10^3^ cfu/mL	Hot embossing	[191]
Biosensor PCR	Paper	Bacterial sample	6.3 × 10^−2^ pmol/L	Wax screen printing	[192]
Immuno-magnetic nanoparticles, magnetic changes	Thermoplastic material	Bacterial sample	10 cells per 1 μL	Hot embossing	[194]
LAMP visual detection	PMMA	Food sample (i.e., infant’s formula)	3.8 × 10^2^ cfu/mL	Hot embossing	[195]
LAMP colorimetric test	PMMA	Bacterial sample	10 cells per pathogen	Hot embossing	[196]
LAMP electrochemical sensor	Plastic	Bacterial sample	1.25 pg DNA per reaction	3D printing technique	[197]
Magnetophoresis	Polyethylene PDMS	Milk Cabbage	620 fg (femtograms)	Soft Lithography	[198]
** *Staphylococcus aureus* **					
Biosensor, Colorimetric	Whatman’s No.1 filter paper	Bacterial sample	80 cfu/mL	CO_2_ laser treatment	[202]
(SERS) biosensor, PCR & MLST	SU-8	Clinical samples	17.400 SERS spectra (optofluidic system)	Masking UV exposure	[203]
Immunosensor	PDMS & aluminum foil	Bacterial sample	10^2^ cfu/mL	Oxygen plasma	[202]
(GO) nano biosensor	Paper/Glass/PDMS	Spiked bacterial samples	5.2 × 10^4^ cfu/mL	Soft Lithography	[203]
RPA	PDMS	Bacterial samples	<10 copies	Hot embossing	[204]
LAMP	Polycarbonate (PC)	Clinical samples	20 cfu/reaction	Injection molding	[205]
PCR	PDMS/glass	Food & bacterial sample	-	UV hot embossing	[206]
Plasmon resonance platform, Fluorescence imaging	PMMA	Bacterial sample	-	Laminated object manufacturing (LOM)	[208]
Fluorescence	PDMS SU8-3025	Bacterial sample	1.5 × 10^1^ cfu/ μl	Soft Lithography-silane treatment	[209]
***Campylobacter* spp.**					
Colorimetric	PDMs/glass/PVDF membrane	milk and poultry	1 × 10^2^ cfu/mL 1 × 10^4^ cfu/25 g	Photolithography	[213]
LAMP assay/Colorimetric	Lab on disk from PMMA	Coastal water	7.92 × 10^−3^–9.54 × 10^−1^ pg of genomic DNA/reaction	-	[214]
droplet PCR/color-coded beads	PDMS/SU-8/silicon wafer	Bacterial sample	1 DNA target/droplet	Soft lithography	[215]
Optical biosensor	PMMA/PDMS/glass/OPD	real water samples	1 × 10^5^ cells/mL for *C. jejuni*	Injection molding	[217]

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
