# Peer review of "Polymeric and Paper-Based Lab-on-a-Chip Devices in Food Safety: A Review"

_micromachines, 2023, doi:10.3390/mi14050986_

Round 1
Reviewer 1 Report
Some concerns need to be addressed before publication.
1. Figure 1 is blurry.
2. As a review paper for microfluidics in food safety, the relative amount of papers published in recent years should be presented as a chart.
3. Table 3 should be moved to part 3. A table in the conclusion session is rarely seen.
4. In the conclusion session, a more detailed discussion could be helpful. E.g., from the aspect of the fabrication method, detection method, and new material.
5. More comparisons between polymer and paper should be provided: the advantage and disadvantages of polymer-based and paper-based material.
6. Besides the discussion on the applications of the microfluidic devices, as a tool for food analysis, the antibacterial properties of the microfluidic chip should also be briefly discussed.
Author Response
Our detailed response is inside the attached document.
Thank you for your valuable comments.

Reviewer 2 Report
There is much interest in lab-on-a-chip devices for food safety, and the topic of the review dedicated to this subject is of interest to readers. In this sense, this review is highly commendable because it is responsive to this interest. The contents are scientifically appropriate and the issues are well organized. Chapter 3 in particular is well explained, including the latest topics. However, the method of preparation in Chapter 2 is too verbose, and I think that many of the points do not need to be explained again in this review. Therefore, it would be better to list the requirements for the methods commonly used in food testing and briefly explain the two most commonly used methods, the paper type and the polymer type. I think it is necessary to revise this point.
The English expressions are sufficient for understanding the content.
Author Response
Thank you for your positive judgement. We have also addressed your comment on section 2. On the attached file you can find our detailed response.
